# Whole genome analysis of a schistosomiasis-transmitting freshwater snail

Coen M. Adema et al.[#]

*Biomphalaria* snails are instrumental in transmission of the human blood fluke *Schistosoma mansoni*. With the World Health Organization's goal to eliminate schistosomiasis as a global health problem by 2025, there is now renewed emphasis on snail control. Here, we characterize the genome of *Biomphalaria glabrata*, a lophotrochozoan protostome, and provide timely and important information on snail biology. We describe aspects of phero-perception, stress responses, immune function and regulation of gene expression that support the persistence of *B. glabrata* in the field and may define this species as a suitable snail host for *S. mansoni*. We identify several potential targets for developing novel control measures aimed at reducing snail-mediated transmission of schistosomiasis.

#A full list of authors and their affiliations appears at the end of the paper.

The fresh water snail *Biomphalaria glabrata* (Lophotrochozoa, Mollusca) is of medical relevance as this Neotropical gastropod contributes as intermediate host of *Schistosoma mansoni* (Lophotrochozoa, Platyhelminthes) to transmission of the neglected tropical disease human intestinal schistosomiasis[1]. Penetration by an *S. mansoni* miracidium into *B. glabrata* initiates a chronic infection in which the parasite alters snail neurophysiology, metabolism, immunity and causes parasitic castration such that *B. glabrata* does not reproduce but instead supports generation of cercariae, the human-infective stage of *S. mansoni*. The complex molecular underpinnings of this long term, intimate parasite-host association remain to be fully understood. Patently infected snails release free-swimming cercariae that penetrate the skin of humans that they encounter in their aquatic environment. Inside the human host, *S. mansoni* matures to adult worms that reproduce sexually in the venous system surrounding the intestines, releasing eggs, many of which pass through the intestinal wall and are deposited in water with the feces. Miracidia hatch from the eggs and infect another *B. glabrata* to complete the life cycle. Related *Biomphalaria* species transmit *S. mansoni* in Africa. Schistosomiasis is chronically debilitating. Estimates of disease burden indicate that disability-adjusted life years lost due to morbidity rank schistosomiasis second only to malaria among parasitic diseases in impact on global human health[2].

In the absence of a vaccine, control measures emphasize mass drug administration of praziquantel (PZQ), the only drug available for large-scale treatment of schistosomiasis[3]. Schistosomes, however, may develop resistance and reduce the effectiveness of PZQ[4]. Importantly, PZQ treatment does not protect against re-infection by water-borne cercariae released from infected snails. Snail-mediated parasite transmission must be interrupted to achieve long-term sustainable control of schistosomiasis[5]. The World Health Organization has set a strategy that recognizes both mass drug administration and targeting of the snail intermediate host as crucial towards achieving global elimination of schistosomiasis as a public health threat by the year 2025 (ref. 6). This significant goal provides added impetus for detailed study of the biology of *B. glabrata*.

Here we characterize the *B. glabrata* genome and describe biological properties that likely afford the snail's persistence in the field, a prerequisite for schistosome transmission, and that may shape *B. glabrata*/*S. mansoni* interactions, including aspects of immunity and gene regulation. These efforts, we anticipate, will foster developments to interrupt snail-mediated parasite transmission in support of schistosomiasis elimination.

## Results

**Genome sequencing and analysis.** The *B. glabrata* genome has an estimated size of 916 Mb (ref. 7) and comprises eighteen chromosomes (Supplementary Figs 1–3; Supplementary Note 1). We assembled the genome of BB02 strain *B. glabrata*[8] (~78.5× coverage) from Sanger sequences (end reads from ~136 kbp BAC inserts[8]), 454 sequences (short fragments, mate pairs at 3 and 8 kbp) and Illumina paired ends (300 bp fragments; Supplementary Data 1). Automated prediction (Maker 2)[9] yielded 14,423 gene models (Methods). A linkage map was used to assign genomic scaffolds to linkage groups (Supplementary Note 2; Supplementary Data 2). We mapped transcriptomes (Illumina PE reads) from 12 different tissues of BB02 snails (Methods; Supplementary Data 1) onto the assembly to aid gene annotation. The pile up of reads revealed polymorphic transcripts (containing single nucleotide variants; SNV), that were correlated through KEGG[10] analyses with metabolic pathways represented in the predicted proteome and the secretome (Supplementary Figs 4–7; Supplementary Note 3; Supplementary Data 7–8). Combined with delineation of organ-specific patterns of gene expression (Supplementary Figs 8 and 9; Supplementary Note 4; Supplementary Data 9), this provided potential molecular markers to help interpret *B. glabrata*'s responses to environmental insults and pathogens, including schistosome-susceptible mechanisms and resistant phenotypes.

**Communication in an aquatic environment.** Aquatic molluscs employ proteins for communication; for example, *Aplysia* attracts conspecifics using water-soluble peptide pheromones[11]. We collected *B. glabrata* proteins from snail conditioned water (SCW) and following electrostimulation (ES), which induces rapid release of proteins. The detection by NanoHPLC-MS/MS of an orthologue of temptin, a pheromone of *Aplysia*[12], among these proteins (Supplementary Note 5; Supplementary Data 10) suggests an operational pheromone sensory system in *B. glabrata*. To explore mechanisms for chemosensory perception, the *B. glabrata* genome was analysed for candidate chemosensory receptor genes of the G-protein-coupled receptor (GPCR) superfamily. We identified 241 seven transmembrane domain GPCR-like genes belonging to fourteen subfamilies, that cluster in the genome. RT–PCR and *in situ* hybridization confirmed expression of a GPCR-like gene within *B. glabrata* tentacles, known to be involved in chemosensation (Fig. 1). Use of chemical communication systems to interact with conspecifics may have a tradeoff effect by potentially exposing *B. glabrata* as a target for parasites (Supplementary Figs 10 and 11; Supplementary Note 6; Supplementary Data 11) and that can be developed to interfere with snail mate finding and/or host location by parasites.

**Stress and immunity.** To persist in the environment, *B. glabrata* must manage diverse stressors, including heat, drought, xenobiotics, pollutants and pathogens including *S. mansoni*. Additional to previous reports of *Capsaspora*[13] a single-cell eukaryote endosymbiont, we noted from the sequenced material an unclassified mycoplasma (or mollicute bacteria) and viruses (Supplementary Figs 12 and 13; Supplementary Notes 7 and 8; Supplementary Data 12). Pending further characterization of prevalence, specificity of association with *B. glabrata,* and impact on snail biology, these novel agents may find application in genetic modification of *B. glabrata* or control of snails through use of specific natural pathogens. Five families of heat-shock proteins (HSP): HSP20, HSP40, HSP60, HSP70 and HSP90 contribute to anti-stress response capabilities of *B. glabrata*. The *HSP70* gene family is the largest with six multi-exon genes, five single-exon genes, and over ten pseudogenes (Supplementary Figs 14–17; Supplementary Note 9; Supplementary Data 13). In general, it is anticipated that future genome assemblies and continued annotation efforts can identify additional *B. glabrata* genes and provide updated gene models to reveal that some current pseudogenes are in fact intact functional genes. The existence of a single-exon *HSP70* gene, however, was independently confirmed by sequence obtained from *B. glabrata* BAC clone (BG_BBa-117G16, Genbank AC233578, basepair interval 49686-51425) and this supports the notion that prediction of single exon gene models for several *HSP70* genes from the current genome assembly is accurate. Retention of *HSP* genes in *B. glabrata* embryonic (*Bge*) cells, the only available molluscan cell line[14], enables *in vitro* investigation of anti-stress and pathogen responses involving *B. glabrata* HSPs (Supplementary Figs 18–22; Supplementary Note 10; Supplementary Data 14). In addition, *B. glabrata* has about 99 genes encoding haem-thiolate enzymes (CYP superfamily)

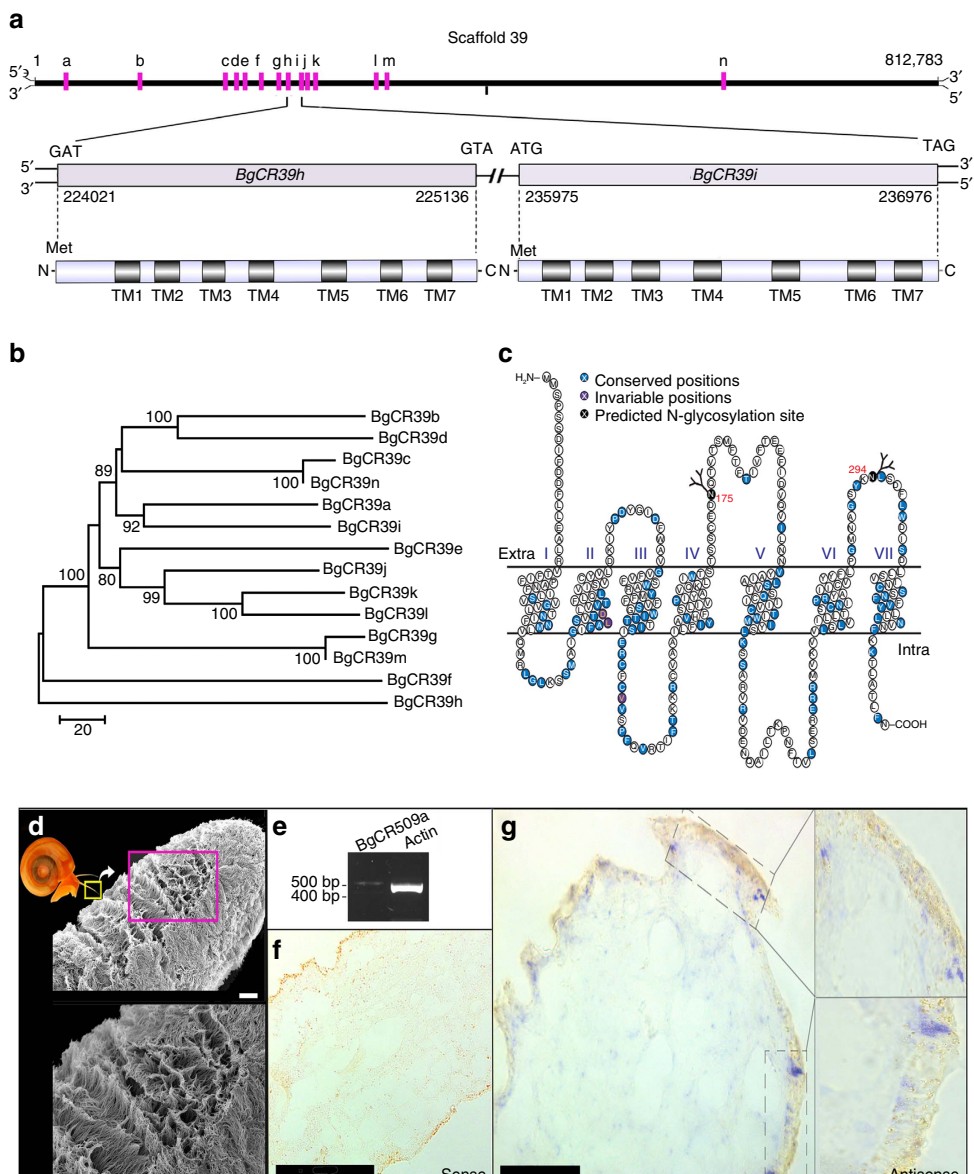

**Figure 1 | Candidate chemosensory receptors of *B. glabrata*. (a)** LGUN_random_Scaffold39 contains fourteen candidate chemosensory receptor (CR) genes (BgCRa-n). Most encode seven-transmembrane domain G-protein-coupled receptor-like proteins, BgCRm and BgCRn are truncated to six-transmembrane domains. See Supplementary Data 11 for gene model identifiers. **(b)** Phylogenetic analysis (neighbour joining, scale bar represents amino-acid substitutions per site) of chemosensory receptors on LGUN_random_Scaffold39 (protein-level). **(c)** Schematic of receptor showing conserved and invariable amino acids, transmembrane domains I-VII; and location of glycosylation sites. **(d)** Scanning electron micrograph showing anterior tentacle, with cilia covering the surface. Scale bar, 20 μm (top); 10 μm (bottom). **(e)** RT–PCR gel showing amplicon for BgCR509a and actin from *B. glabrata* tentacle. **(f,g)** *In situ* hybridization showing sense (negative control) and antisense localization of BgCR509a mRNA in anterior tentacle section (purple). Scale bar (**f**): 100 μm; (**g**) 50 μm.

toward detoxifying xenobiotics, with representation of all major animal cytochrome P450 clans. Eighteen genes of the mitochondrial clan suggest that molluscs, like arthropods, but unlike vertebrates, also utilize mitochondrial P450s for detoxification[15]. Tissue-specific expression (for example, four transcript sequences uniquely in ovotestis) suggests that 15 *P450* genes serve specific biological processes. These findings indicate potential for rational design of selective molluscicides, for example, by inhibiting unique P450s or by activation of the molluscicide only by *B. glabrata*-specific P450s (Supplementary Note 11; Supplementary Data 16).

*Biomphalaria glabrata* employs pattern recognition receptors (PRRs)[16] to detect pathogens and regulate immune responses.

These include 56 *Toll-like receptor* (*TLR*) genes, of which 27 encode complete TLRs (Fig. 2; Supplementary Note 12; Supplementary Data 17), associated with a signaling network for transcriptional regulation through NF-κB transcription factors (Supplementary Fig. 23; Supplementary Note 13; Supplementary Data 18). Like other lophotrochozoans, *B. glabrata* shows expansion of *TLR* genes relative to mammals and insects which have ~10 TLRs[17]. Other PRRs include eight peptidoglycan recognition-binding proteins (PGRPs), and a single Gram-negative binding protein (GNBP; Supplementary Note 12; Supplementary Data 17). A prominent category of *B. glabrata* PRRs consists of fibrinogen-related proteins (FREPs), plasma lectins that are somatically mutated to yield unique FREP

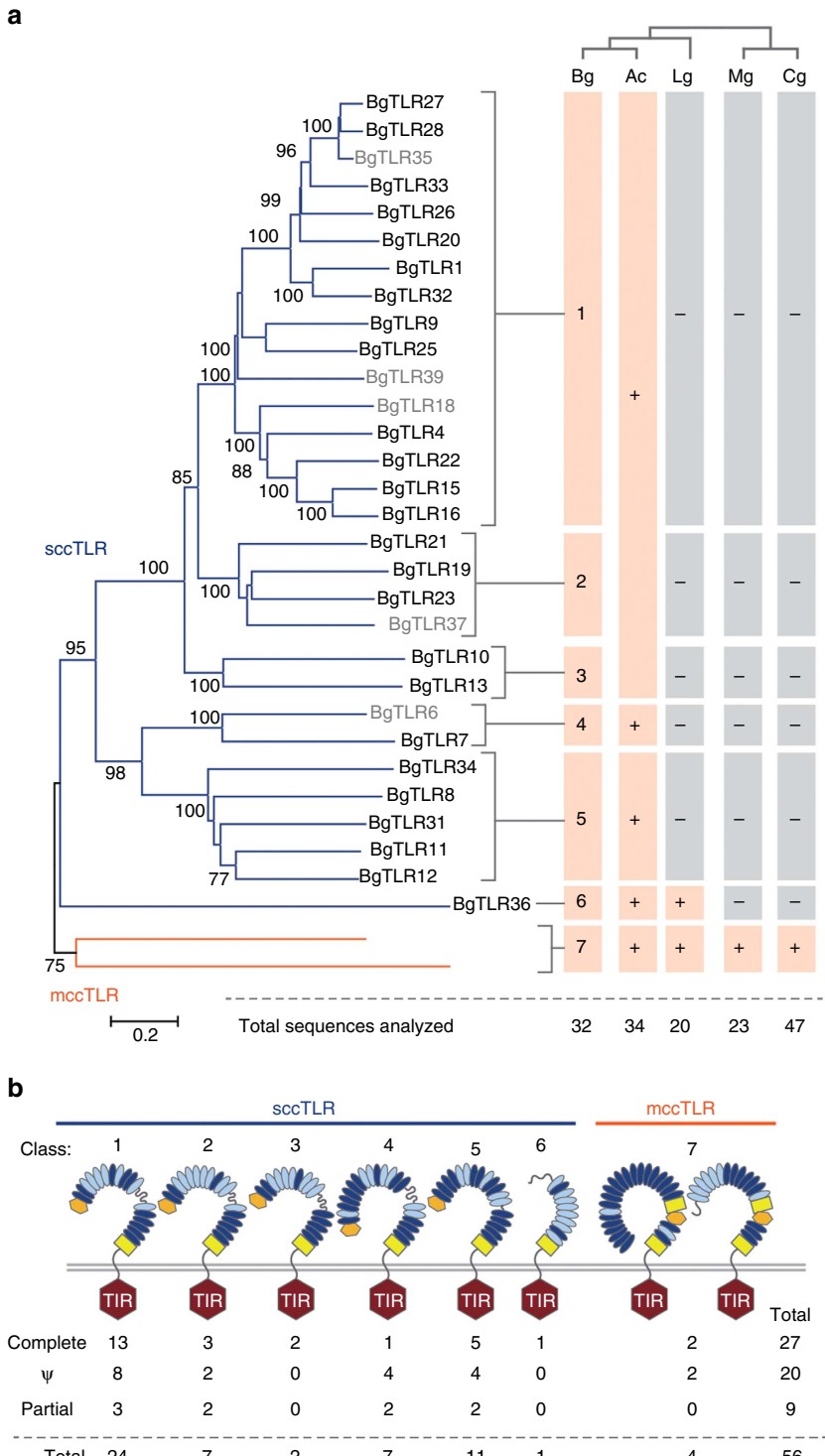

**Figure 2 | TLR genes in *B. glabrata*.** (**a**) Analysis of the (complete) TIR domains from BgTLRs identified seven classes (neighbour-joining tree, scale bar represents amino-acid substitutions per site). Bootstrap values shown for 1,000 replicates. Comparisons included TLRs from *A. californica* (Ac), *L. gigantea* (Lg), *Mytilus galloprovincialis* (Mg) and *C. gigas* (Cg). The presence or absence of orthologues of each class in each molluscan species is indicated. A representative of the *B. glabrata* class 1/2/3 clade is present within *A. californica*, but is independent of the *B. glabrata* TLR classes (indicated by the large pink box). Grey font indicates pseudogenes or partial genes. (**b**) *B. glabrata* has both single cysteine cluster (scc; blue line)- and multiple cysteine cluster (mcc; orange line) TLRs. Domain structures are shown for BgTLR classes. BgTLRs consist of an LRRNT (orange hexagon), a series of LRRs (ovals), a variable region (curvy line), LRRCT (yellow box), and transmembrane domain, and an intracellular TIR domain (hexagon). The dark blue ovals indicate well defined LRRs (predicted by LRRfinder[57]); light blue ovals are less confident predictions. Each of the two class 7 BgTLRs has a distinct ectodomain structure. The numbers of complete, pseudogenes (Ψ) and partial genes are indicated for each class.

repertoires in individual snails[18]. Our analyses revealed that this PRR diversity is generated from a limited set of germline sequences comprising 20 FREP genes with two upstream IgSF domains preceding an fibrinogen (FBG)-like domain, and four FREP genes encoding one immunoglobulin (IgSF) domain and one C-terminal FBG-like domain, including one gene with an N-terminal PAN_AP domain. FREP genes cluster in the genome, often accompanied by partial FREP-like sequences (Supplementary Figs 24–27; Supplementary Note 14). A proteomics level study indicated that S. mansoni resistance in B. glabrata associates with expression of parasite-binding FREPs of particular gene families, as well as GREP (galectin-related protein), FREP-like lectins that instead of a C-terminal FBG domain contain a galectin domain[19] (Supplementary Figs 18,19; Supplementary Note 10; Supplementary Data 15). Further analyses yielded novel aspects of B. glabrata immune capabilities. We identified several cytokines, including twelve homologs of IL-17, four MIF homologs, and eleven TNF sequences (Supplementary Note 12; Supplementary Data 17). Biomphalaria glabrata possesses gene orthologs of complement factors that may function to opsonize pathogens (Supplementary Figs 28 and 29; Supplementary Note 15, Supplementary Data 19).

We discovered an extensive gene set for apoptosis, a response that can regulate invertebrate immune defense[20], including ∼50 genes encoding for Baculovirus IAP Repeat (BIR) domain-containing caspase inhibitors. The expansion of this gene family in molluscs (17 genes in Lottia gigantea, 48 in Crassostrea gigas), relative to other animal clades, suggests important regulatory roles in apoptosis and innate immune responses of molluscs[21] (Supplementary Figs 30–32; Supplementary Note 16; Supplementary Data 20). We characterized a large gene complement to metabolize reactive oxygen species (ROS) and nitric oxide (NO) that are generated by B. glabrata hemocytes to exert cell-mediated cytotoxicity toward pathogens, including schistosomes (Supplementary Fig. 33; Supplementary Note 17; Supplementary Data 21).

The antimicrobial peptide (AMP) arsenal of B. glabrata is surprisingly reduced compared to other invertebrate species (for example, bivalve molluscs have multiple AMP gene families[22]); our searches indicated only a single macin-type gene family, comprising six biomphamacin genes. However, B. glabrata does possess multigenic families of antibacterial proteins including two achacins, five LBP/BPIs, and 21 biomphalysins (Supplementary Fig. 34; Supplementary Note 18; Supplementary Data 22 and 23). While gaps in functional annotation limit our interpretation of B. glabrata immune function (Supplementary Note 19; Supplementary Data 24 and 25), our analyses reveal a multifaceted, complex internal defense system that must be evaded or negated by parasites such as S. mansoni to successfully establish infection.

**Regulation of biological processes**. Characterization of the regulatory mechanisms that rule gene expression and general biological functions is especially interesting because survival of B. glabrata relies on the capacity to quickly recognize, respond, and adapt to external and internal signals. In addition, a better understanding of parasite–host compatibility will be afforded by characterization of snail control mechanisms for gene expression and signalling pathways as possible targets for interference by S. mansoni to alter host physiology, including reproductive activities, to survive in B. glabrata[23]. Gene expression in B. glabrata is under epigenetic regulation[24–26], we identified chromatin-modifying enzymes including class I and II histone methyltransferases, LSD-class and Jumonji-class histone demethylases, class I–IV histone deacetylases, and GNAT, Myst and CBP superfamilies of histone acetyltransferases.

Biomphalaria has homologues of DNA (cytosine-5)-methyltransferases 1 and 2 (no homolog of DNMT3), as well as putative methyl-CpG binding domain proteins 2/3. In silico analyses predicted a mosaic type of DNA methylation, as is typical for invertebrates (Supplementary Figs 35–39; Supplementary Note 20; Supplementary Data 26). The potential role of DNA methylation in B. glabrata reproduction and S. mansoni interactions is reported in a companion paper[27].

The B. glabrata genome also encodes the protein machinery for biogenesis of microRNA (miRNAs) to regulate gene expression (Supplementary Note 21; Supplementary Data 27). Two computational methods independently predicted the same 95 pre-miRNAs, encoding 102 mature miRNAs. Of these, 36 miRNAs were observed within our transcriptome data, another 53 miRNAs displayed ≥90% nucleotide identity with L. gigantea miRNAs. Bioinformatics predicted 107 novel pre-miRNAs unique to B. glabrata. Based on the analysis of binding thermodynamics and miRNA:mRNA structural features, several novel miRNAs were predicted to likely regulate transcripts involved in processes unique to snail biology, including secretory mucosal proteins and shell formation (biomineralization) that may present possible targets for control of B. glabrata (Supplementary Figs 40–67; Supplementary Note 21 and 22; Supplementary Data 28–33).

Periodicity of aspects of B. glabrata biology[28] indicates likely control by circadian timing mechanisms. We identified seven candidate clock genes in silico, including a gene with strong similarity to the period gene of A. californica. Modification of expression of clock genes may interrupt circadian rhythms of B. glabrata and affect feeding, egg-laying and emergence of cercariae (Supplementary Note 23).

Neuropeptides expressed within the nervous system coordinate the complex physiology of B. glabrata, a simultaneous hermaphrodite snail. In silico searches identified 43 B. glabrata neuropeptide precursors, predicted to yield over 250 mature signalling products. Neuropeptide transcripts occurred in multiple tissues, yet some were most prominent within terminal genitalia (49%) and the CNS (56%), or even specific to the CNS, including gonadotropin-releasing hormone (GnRH) and insulin-like peptides 2 and 3 (Supplementary Fig. 68; Supplementary Note 24; Supplementary Data 34–36). The reproductive physiology of hermaphroditic snails is also modulated by male accessory gland proteins (ACPs), which are delivered with spermatozoa to augment fertilization success[29]. The B. glabrata genome has sequences matching one such protein, Ovipostatin (LyAcp10), but none of the other ACPs identified in Lymnaea stagnalis[30]. Putatively, ACPs evolve rapidly and are taxon specific (Supplementary Fig. 69; Supplementary Note 25; Supplementary Data 34), such that they allow for specific targeting of reproductive activity for control measures.

A role of steroid hormones in reproduction of hermaphrodite snails with male and female reproductive organs remains speculative. Biomphalaria glabrata has a CYP51 gene to biosynthesize sterols de novo, yet we found no orthologs of genes involved in either vertebrate steroid or arthropod ecdysteroid biosynthesis. The lack of CYP11A1 suggests that B. glabrata cannot process cholesterol to make vertebrate-like steroids. The absence of aromatase (CYP19), required for the formation of estrogens, is particularly enigmatic as molluscs possess homologues of mammalian estrogen receptors. Characterization of snail-specific aspects of steroidogenesis may identify targets to disrupt reproduction towards control of snails. (Supplementary Fig. 70; Supplementary Note 26; Supplementary Data 37).

Eukaryotic protein kinases (ePKs) and phosphatases constitute the core of cellular signaling pathways, playing a central role in signal transduction by catalyzing reversible protein

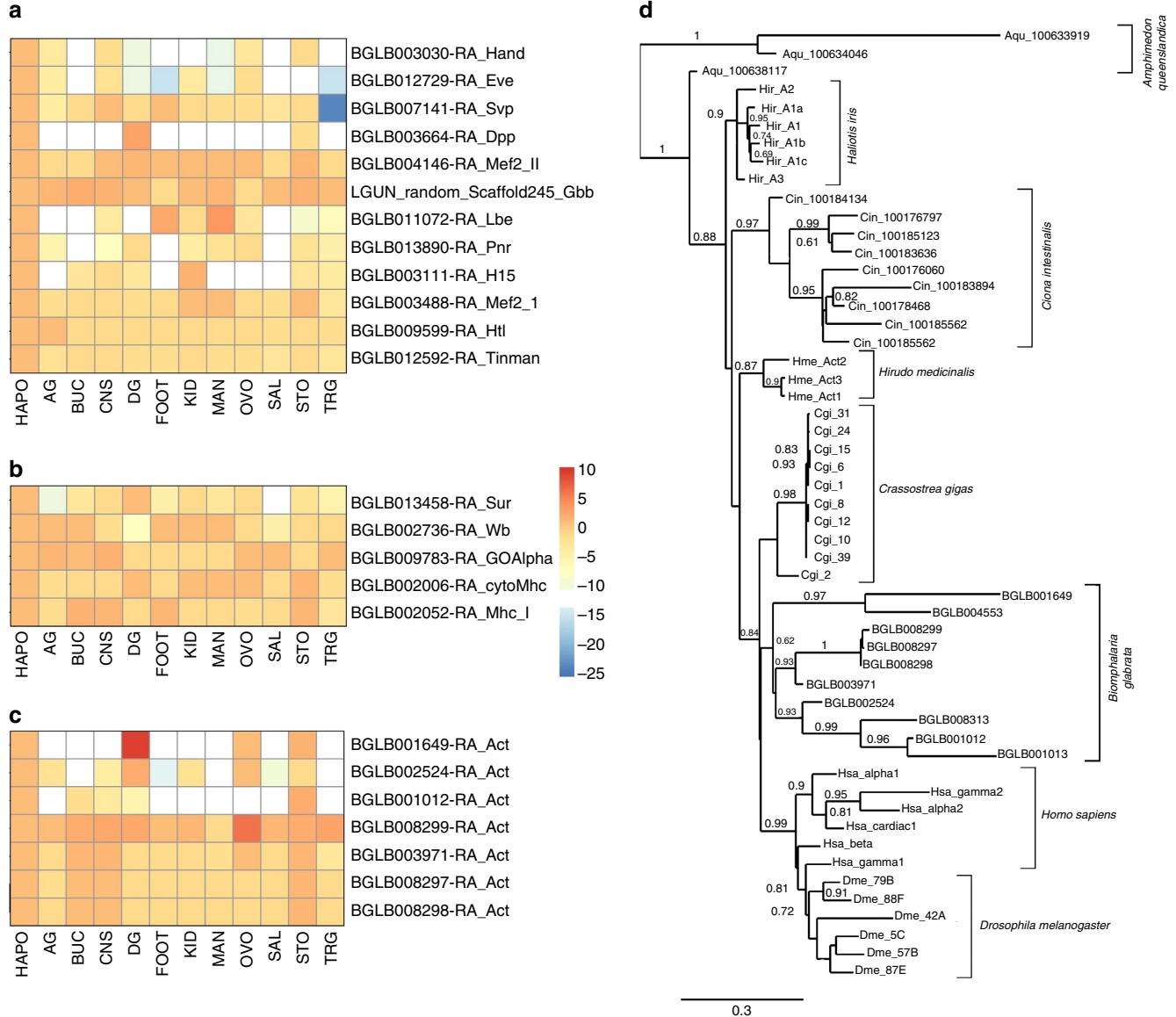

**Figure 3 | Expression of cardiac genes and actin genes in *B. glabrata* tissues.** (**a**) Cardiac regulatory genes. (**b**) Cardiac structural genes. (**c**) Relative expression of actin genes in *B. glabrata* tissues. For (**a**–**c**), the score represents gene level aggregate of normalized FPKM counts for *de novo* assembled tissue transcripts, relative to expression levels in the heart/APO sample. The counts were scaled (with median read count as 0) to indicate expression intensity with red indicating highest, blue lowest. AG, Albumen gland; BUC, buccal mass; CNS, central nervous system; DG, digestive gland; FOOT, headfoot; HAPO, heart/APO; KID, kidney; MAN, mantle edge; OVO, ovotestes; SAL, salivary glands; STO, stomach; TRG, terminal genitalia. (**d**) Maximum Likelihood tree (Phylogeny.fr, scale bar represents amino-acid substitutions per site) showing phylogenetic relationships of actin genes, based on amino-acid sequence alignment (ClustalW). *Biomphalaria* -snail; *Crassostrea gigas*—oyster; *Haliotis iris*- abalone;, *Hirudo medicinalis* – leech (all lophotrochozoans); *Amphimedon queenslandica*, sponge, Prebilateria, ophotrochozoans), *Drosophila melanogaster*—fruit fly, Ecdysozoa), and the deuterostomes *Ciona intestinalis*, sea squirt; *Homo sapiens*, human. See Supplementary Note 31 for accession numbers.

phosphorylation in non-linearly integrated networks. *Schistosoma mansoni* likely interferes with the extracellular signal-regulated kinase (ERK) pathway to survive in *B. glabrata*[23]. Hidden Markov model searches on the predicted *B. glabrata* proteome identified 240 potential ePKs, encompassing all main types of animal ePKs (Supplementary Fig. 71; Supplementary Note 27). Similarity searches also identified 60 putative protein phosphatases comprising ~36 protein Tyr phosphatases (PTPs) and ~24 protein Ser/Thr phosphatases (PSPs) (Supplementary Figs 72–74; Supplementary Note 28). These sequences can be studied for understanding control of homeostasis, particularly in the face of environmental and pathogenic insults encountered by *B. glabrata*.

**Bilaterian evolution.** Genome study of *B. glabrata* can also provide new insights into evolution of bilaterian metazoa by increasing diversity of the relatively few lophotrochozoan taxa that have been characterized to date (that is, platyhelminths, leech, bivalve, cephalopod and polychaete)[31–35]. Comparison of similar biological features and gene expression patterns among lophotrochozoans, ecdysozoans and deuterostomes may indicate the evolutionary origin of conserved gene families and anatomical features. The prevalence in diverse taxa of metazoa, including molluscs, arthropods and chordates, of muscular heart-like organs that function to circulate blood or hemolymph, has led to the proposal that these structures evolved over evolutionary time from a primitive heart present in an urbilaterian ancestor.

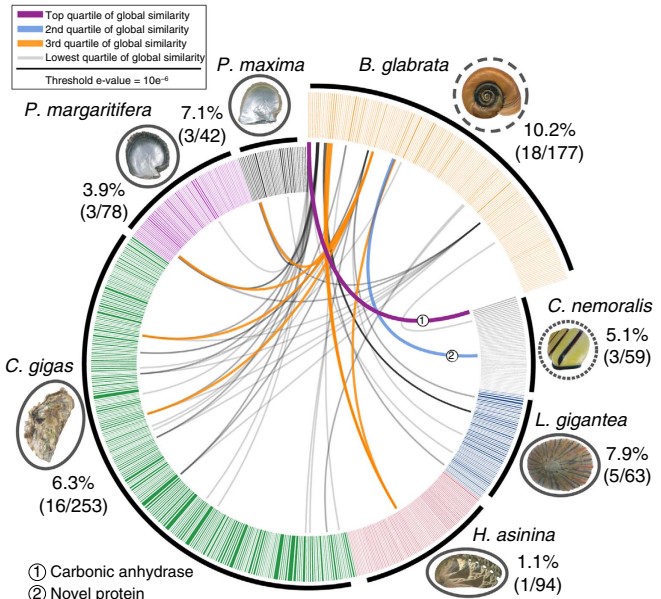

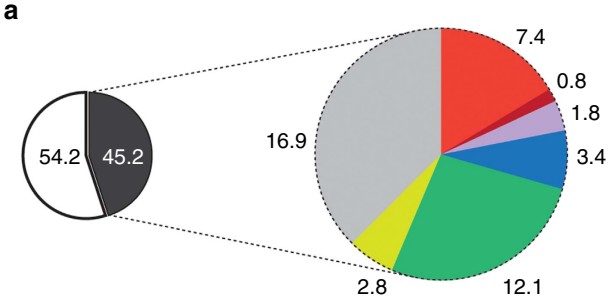

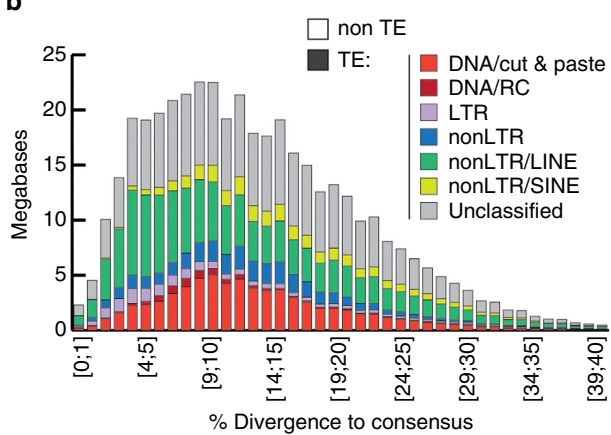

**Figure 4 | Comparison of molluscan shell forming proteomes.** Circos diagram of 177 mantle-specific, secreted *Biomphalaria* gene products compared against shell forming proteomes of six other molluscs (BLASTp threshold ≤10e⁻⁶). Protein pairs that share sequence similarity in the top quartile are linked in purple, the second quartile is linked in blue, third quartile has orange links and lowest quartile of similarity has grey links. Species that occupy marine habitats are surrounded by a solid line, A finely dashed line identifies the terrestrial species *Cepea nemoralis*, the freshwater species *B. glabrata* has a coarse line. Percentages (and proportions in brackets) indicate the number of proteins that shared similarity with a *Biomphalaria* shell forming candidate gene. The width of each sector line around the ideogram is proportional to the length of that gene in basepairs. Photographs taken by DJ Jackson, with exception of photograph of *C. gigas*, by David Monniaux, distributed under a CC-A SA 3.0 license.

**Figure 5 | Transposable element (TE) landscape of *B. glabrata*.**
(**a**) Left: proportion (%) of the genome assembly annotated as TE (black). Right: TE composition by class (indicating % of the genome corresponding to each class). (**b**) Evolutionary view of TE landscape. For each class, cumulative amounts of DNA (in Mb) are shown as function of the percentage of divergence from the consensus (by bins of 1%, first one being ≥0 and <1; see Supplementary Note 32 for Methods). Percentage of divergence from consensus is used as a proxy for age: the older the invasion of the TE is, the more copies will have accumulated mutations (higher percentage of divergence, right of the graph; left of the graph: youngest elements showing little divergence from consensus). Note that the result of this analysis of assembled sequence does not exclude the likelihood that intact transposable elements are present in *B. glabrata*. Colors are as in **a**. RC, rolling circle.

This hypothesis is supported by similarities in core genes for specification and differentiation of cardiac structures between insects (in particular *Drosophila*) and vertebrates[36,37]. To further develop this notion, we searched for molluscan cardiac-specification and -differentiation genes in the genome of *B. glabrata*. A previously characterized short cDNA sequence from snail heart RNA led to identification of BGLB012592 as the *Biomphalaria* ortholog of *tin/Nkx2.5* (ref. 38). Similarity searches with *Drosophila* orthologues identified most of the core cardiac regulatory factors and structural genes in the *B. glabrata* genome (Supplementary Note 29; Supplementary Data 38), with enriched expression of these genes in cardiac tissues (Fig. 3). Pending confirmation of functional involvement of these core cardiac genes in heart formation, these results from a lophotrochozoan, in conjunction with ecdysozoans and deuterostomes, merit continued consideration of the presence of a primitive heart-like structure and in the urbilaterian ancestor.

We also investigated in molluscs, relative to insects and mammals, the evolution of the gene family of actins, conserved proteins that function in cell motility (cytoplasmic actins) and muscle contraction (sarcomeric actins)[39]. Previous study showed that cephalopod actin genes[40], are more closely related to one another than to any single mammalian gene, an observation also made another mollusc *Haliotis*[41] and for insect actins[42]. Thus, it has been proposed that actin diversification in arthropods, molluscs and vertebrates each occurred independently. However, it has not been determined whether different molluscan lineages independently underwent actin gene divergence, and few studies

have analysed expression of mollusc actin genes in different tissues[41,43]. We identified ten actin genes in *B. glabrata* that are clustered across seven scaffolds to suggest that some of these genes arose through tandem duplication. Expression across all tissues indicates that four genes encode cytoplasmic actins (Fig. 3). Protein sequence comparisons placed all *B. glabrata* actins as most closely related to mammalian cytoplasmic rather than sarcomeric actins (Supplementary Note 30; Supplementary Data 39), a pattern also observed for all six actin genes of *D. melanogaster*[44]. The actin genes of *B. glabrata* and other molluscs were most similar to paralogs within their own genomes, rather than to other animal orthologs (Fig. 3). One interpretation is that actin genes diverged independently multiple times in molluscs, similar to an earlier hypothesis for independent actin diversification in arthropods and chordates[42]. Alternatively, a stronger appearance of monophyly than really exists may result if selective pressures due to functional constraints keep actin sequences similar within a genome, for example if the encoded proteins have overlapping functions.

To gain insight into the diversification of mechanisms involved in biomineralization in molluscs, we analyzed the transcriptomic data for *B. glabrata* genes involved in biomineralization. Of 1,211

transcripts that were more than twofold upregulated in the mantle relative to other tissues, 34 shared similarity with molluscan sequences known to be involved in shell formation and biomineralization. Another 177 candidate sequences putatively involved in shell formation including 18 genes (10.2%) with similarity to sequences of shell forming secretomes of other marine and terrestrial molluscs were identified from the entire mantle transcriptome (Fig. 4). Highly conserved components of the molluscan shell forming toolkit include carbonic anhydrases and tyrosinases[33] (Supplementary Fig. 75; Supplementary Note 31; Supplementary Data 40). In summary, this genome-level analysis of a subset of molluscan molecular pathways provides new insight into the evolutionary origins of bilaterian organs, gene families and genetic pathways.

**Repetitive landscape**. Repeat content analysis showed that 44.8% of the *B. glabrata* assembly consists of transposable elements (TEs; Fig. 5; Supplementary Figs 76-78; Supplementary Note 32; Supplementary Data 41), comparable to *Octopus bimaculoides* (43%)[34] and higher than observed in other molluscs: Owl limpet, *L. gigantea* (21%)[31]; Pacific oyster, *C. gigas* (36%)[32]; Sea hare, *A. californica* (30%)[45]. The fraction of unclassified elements in *B. glabrata* was high (17.6%). Most abundant classified repeats were LINEs, including Nimbus[46] (27% of TEs, 12.1% of the genome), and DNA TEs (17.7% of TEs, 8% of the genome). Long terminal repeats (LTRs) represented 6% of TEs (1.7% of the genome), and non-mobile simple repeats comprised 2.6% of the genome (with abundant short dinucleotide satellite motifs). Divergence analyses of element copy and consensus sequences indicated that DNA TEs were not recent invaders of the *B. glabrata* genome; no intact transposases were detected in the assembly. A hAT DNA transposon of *B. glabrata* (~1,000 copies) has significant identity with *SPACE INVADERS (SPIN)* which horizontally infiltrated a range of animal species, possibly through host-parasite interactions[47]. Overall, our results reinforce a model in which diverse repeats comprise a large fraction of molluscan genomes.

## Discussion

The genome of the Neotropical freshwater snail *B. glabrata* expands insights into animal biology by further defining the Lophotrochozoan lineage relative to Ecdysozoa and Deuterostomia. An important rationale for genome analysis of *B. glabrata* pertains to its role in transmission of *S. mansoni* in the New World. Most of the world's cases of *S. mansoni* infection, however, occur in sub-Saharan Africa where other *Biomphalaria* species are responsible for transmission, most notably *Biomphalaria pfeifferi*. Likely due to a shared common ancestor, *B. glabrata* provides a good representation of the genomes of African *Biomphalaria* species[48,49]. At least 90% sequence identity was shared among 196 assembled transcripts collected from *B. pfeifferi* (Illumina RNAseq) and the transcriptome of *B. glabrata* (Supplementary Note 33; Supplementary Data 42–43). Accordingly, our analyses of the *B. glabrata* genome likely reveal biological features that define snail species of the genus *Biomphalaria* as effective hosts for transmission of human schistosomiasis. This work provides several inroads for control of *Biomphalaria* snails to reduce risks of schistosome (re)infection of endemic human populations, an important component of the WHO strategy aimed at elimination of the global health risks posed by schistosomiasis[6]. The following are among options that can be considered[50]. The genetic information uncovered may be applied to characterize and track the field distribution of snail populations that differ in effectiveness of parasite transmission. Targeting aspects of pheromone-based

communication among *Biomphalaria* conspecifics may alter the mating dynamics of these snails and perhaps also to interfere with the intermediate host finding of larval schistosomes. Molluscicide design may be tailored to impact unique gene products and mechanisms for gene regulation, reproduction and metabolism toward selective control of *Biomphalaria* snails. Finally, genetic modification of determinants of intermediate host competence may alter schistosome transmission by *Biomphalaria*. In summary, this report provides novel details on the biological properties of *B. glabrata*, including several that may help determine suitability of *B. glabrata* as intermediate host for *S. mansoni*, and points to potential approaches for more effective control efforts against *Biomphalaria* to limit the transmission of schistosomiasis.

## Methods

The genetic material used for sequencing the genome of the hermaphroditic freshwater snail *Biomphalaria glabrata* was derived from three snails of the BB02 strain (shell diameter 8, 10 and 12 mm, respectively), established at the University of New Mexico, USA from a field isolate collected from Minas Gerais, Brazil, 2002 (ref. 8). Using a genome size estimate of 0.9–1 Gb (ref. 7), we sequenced fragments (450 bp read length; $14.08 \times$ coverage) and paired ends from 3 kb long inserts ($8.12 \times$) and 8 kb long inserts ($2.82 \times$) with reads generated on Roche 454 instrumentation, plus $0.06 \times$ from bacterial artificial chromosome (BAC) ends[8] on the ABI3730xl. Reads were assembled using Newbler (v2.6)[51]. Paired end reads from a 300 bp insert library ($53.42 \times$ coverage) were collected using Illumina instrumentation and assembled *de novo* using SOAP (v1.0.5)[52]. The Newbler assembly was merged with the SOAP assembly using GAA[53] (see Supplementary Data 1 for accession numbers of sequence data sets). Redundant contigs in the merged assembly were collapsed and gaps between contigs were closed through iterative rounds of Illumina mate-pair read alignment and extension using custom scripts. We removed from the assembly all contaminating sequences, trimmed vectors (X), and ambiguous bases (N). Short contigs ($\leq 200$ bp) were removed prior to public release. In the creation of the linkage group AGP files, we identified all scaffolds (145 Mb total) that were uniquely placed in a single linkage group (Supplementary Note 2; Supplementary Data 2). Note that because of low marker density, scaffolds could not be ordered or oriented within linkage groups. The final draft assembly (NCBI: ASM45736v1) is comprised of 331,400 scaffolds with an N50 scaffold length of 48 kb and an N50 contig length of 7.3 kb. The assembly spans over 916 Mb (with a coverage of 98%, 899 Mb of sequence with ~17 Mb of estimated gaps). The draft genome sequence of *Biomphalaria glabrata* was aligned with assemblies of *Lottia* and *Aplysia* (http://biology.unm.edu/biomphalaria-genome/synteny.html) and deposited in the DDBJ/EMBL/GenBank database (Accession Number APKA00000000.1). It includes the genomes of an unclassified mollicute (Supplementary Note 7; accession numbers CP013128). The genome assembly was also deposited in Vectorbase[54] (https://www.vectorbase.org/organisms/biomphalaria-glabrata). Computational annotation using Maker2 (ref. 9) yielded 14,423 predicted gene models, including 96.5% of the 458 sequences from the CEGMA core set of eukaryotic genes[55]. Total RNA was extracted from 12 different tissues/organs dissected from several individual adult BB02 *B. glabrata* snails (shell diameter 10–12 mm; between 2 and 10 snails per sample to obtain sufficient amounts of RNA). RNA was reverse transcribed using random priming, no size selection was done. Illumina RNAseq (paired ends) was used to generate tissue-specific transcriptomes for albumen gland (AG); buccal mass (BUC); central nervous system (CNS); digestive gland/hepatopancreas (DG/HP); muscular part of the headfoot (FOOT); heart including amebocyte producing organ (HAPO); kidney (KID); mantle edge (MAN); ovotestis (OVO); salivary gland (SAL); stomach (STO); terminal genitalia (TRG), see Supplementary Data 1 for accession numbers of sequence data sets. RNAseq data were mapped to the genome assembly (Supplementary Note 3). No formal effort was made to use the RNA-data to systematically enhance the structural annotation. VectorBase did, however, make this RNAseq data available in WebApollo[56] such that the community could use these data to correct exon-intron junctions, UTRs, etc. through community annotation. All of these community-based updates have been incorporated and are available via the current VectorBase gene set. Repeat features were analyzed and masked (Supplementary Note 32; see Vectorbase Biomphalaria-glabrata-BB02_REPEATS.lib, Biomphalaria-glabrata-BB02_REPEATFEATURES_BglaB1.gff3.gz). Further methods and results are described in the Supplementary Information.

**Data availability**. The sequence data that support the findings of this study have been deposited in GenBank with the accession codes SRX005826, -27, -28; SRX008161, -2; SRX648260, -61, -62, -63, -64, -65, -66, -67, -68, -69, -70, -71; SRA480937; SRA480939; SRA480940; SRA480945; TI accessions2091872204-2092480271; 2104228958-2104243968; 2110153721-2118515136; 2181062043-2181066224; 2193113537-2193116528; 2204642410-2204763511; 2204820860-2204852286; 2213009530-2213057324; 2260448774-2260450167. Also see Supplementary Data 1. The assembly and related data are available from

VectorBase, https://www.vectorbase.org/organisms/biomphalaria-glabrata. The *Biomphalaria glabrata* genome project has been deposited at DDBJ/EMBL/GenBank under the accession number APKA00000000.1

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

## Acknowledgements

We thank S. Newfeld for discussion of actin evolution; N. El Sayed and H. Tettelin for discussion of HSP annotation and expression. We acknowledge access to the Metafer microscopy system at the I. Robinson Research Complex, Harwell, Rutherford Appleton Laboratory, Oxon, UK (BBSRC Professorial Fellowship, grant number BB/H022597/1). Sequence characterization of the Biomphalaria glabrata genome was funded by NIH-NHGRI grant HG003079 to R.K.W., McDonnell Genome Institute, Washington University School of Medicine. Biomphalaria glabrata and Schistosoma mansoni were provided to some participating labs by the NIAID Schistosomiasis Resource Center (Biomedical Research Institute, Rockville, MD) through NIH-NIAID Contract HHSN272201000005I for distribution through BEI Resources. C.M.A. and E.S.L. acknowledge NIH grant P30GM110907 from the National Institute of General Medical Sciences (NIGMS). Publication costs were contributed equally by McDonnell Genome Institute, Washington University School of Medicine and the COBRE Center for Evolutionary and Theoretical Immunology (CETI) which is supported by NIH grant P30GM110907 from the National Institute of General Medical Sciences (NIGMS). E.S.L. acknowledges NIH/NIAID ROI AI101438. J.M.B., H.D.A.-G and M.K. acknowledge NIH-NIAID R01-AI0634808. M.Y. acknowledges UK BBSRC (BB/H022597/1). G.O. acknowledges support from FAPEMIG (RED-00014-14, PPM-00189-13) and CNPq (304138/2014-2, 309312/2012-4). R.L.C. acknowledges CNPq (503275/2011-5). T.P.Y. acknowledges NIH/NIAID RO1AI015503. K.F.H. and M.T.S. acknowledge BBSRC (BB/K005448/1). B.G. acknowledges ANR JCJC INVIMORY (ANR-13-JSV7-0009). S.E. acknowledges NIAID contract HHSN272201400029C. J.M.K. acknowledges the Research Council for Earth and Life Sciences (A.L.W.; 819.01.007) and the Netherlands Organization for Scientific Research (NWO). R.M.C. acknowledges NIH GM061738 and support from the American Heart Association, Southwest Affiliate (14GRNT20490250). D.T. acknowledges NIH R25 GM075149. C.F. acknowledges NIH R01-GM077582. D.J.J. acknowledges D.F.G. JA2108/1-2. M.de S.G. acknowledges CNPq 479890/2013-7. K.M.B. and J.P.R. acknowledge NSERC 312221 and CIHR MOP74667. C.S.J., L.R.N., S.J., E.J.R., S.K. and A.E.L. acknowledge NC3R GO900802/1. K.K.B. and K.B.S. acknowledge NSERC 315051 and 6793, respectively. M.B. and C.J.B. acknowledge NIH RO1-AI109134. C.J.B. acknowledges NIH AI016137 and AI111201. B.R. acknowledges NHGRI 4U41HG002371. P.C.H., M.A.G. and E.A.P. acknowledge NSERC 418540. O.L.B. and Ch.C. acknowledge ANR-12-EMMA-0007-01. S.F.C. acknowledges Australian Research Council FT110100990.

## Author contributions

C.M.A., E.S.L., M.K., N.R. conceived the study, scientific objectives. C.M.A. led the project and manuscript preparation with input from steering committee members M.K., C.S.J., G.O., P.M., L.W.H., A.S., E.S.L., assisted by S.E. O.C. provided field collected snails. C.M.A. and E.S.L. cultured snails and provided materials. P.M., L.W.H., S.C., L.F., W.C.W., R.K.W., V.M., C.T. developed the sequencing strategy, managed the project, conducted assembly and evaluation. B.R. performed genome alignments. M.C., D.H., S.E., G.G.-C. and D.L. performed the genebuild, managed metadata, performed genome annotation and data analysis, and facilitated Community Annotation with M.C.M.-T. H.D. A.-G., M.Y., E.V.V., M.K. and J.M.B. performed karyotyping and FISH analysis. J.A.T. and M.B. performed linkage mapping. G.O., J.G.A., Y.C.-A., S.G.G., F.L. F.S.O., F.S.P, I.C.R. and L.L.S.S. performed computational analyses of genomic, proteomic, and transcriptomic data, SNP content, secretome, metabolic pathways and annotation of eukaryote protein kinases (ePKs). S.F.C., L.Y., D.L., M.Z. and D.McM. conducted pheroreception studies. M.T.S., K.K.G., U.N. and K.F.H. conducted bacterial symbiont analysis. S.L., S.-M.Z., E.S.L. and B.C.B. performed virus analyses. M.K., P.F., W.I. and N.R. performed annotation of HSP. T.P.Y., X.-J.W., U.B.-W. and N.D. conducted proteogenomic studies of B. glabrata embryonic (Bge) cells and parasite-reactive snail host proteins and data analysis. R.F., A.E.L. and C.S.J. performed annotation of CYP. J.H. performed annotation of NFkB. K.M.B. and J.P.R. performed annotation of conserved immune factors. C.M.A. and J.J.P. performed annotation of FREPs. M.C. and C.M. performed annotation of complement. D.D. performed annotation of apoptosis. B.G. and C.J.B. performed annotation of REDOX balance. O.L.B., D.D., R.G., Ch.C. and G.M. performed annotation of antibacterial defenses. L.D.S. and A.T.P. performed search for antibacterial defense genes. P.C.H., M.A.G. and E.A.P. performed annotation of unknown novel sequences. K.K.G., I.W.C., U.N., K.F.H., M.T.S., Ce.C., T.Q. and C.G. performed annotation and analysis of epigenetic sequences. E.H.B., L.R. doA., M.de.S.G., R.L.C., and W.de.J.J. performed annotation of miRNA (Brazil), K.K.B., R.P. and K.B.S. performed annotation of miRNA (Canada). M.G. performed annotation of periodicity. S.F.C., B.R., T.W., A.E.L. and S.K. conducted neuropeptide studies and data analysis. A.E.L., R.F., S.K., E.J.R., S.J., D.R., C.S.J. and L.R.N. performed annotation of steroidogenesis. J.M.K., B.R. and S.F.C. performed annotation of ovipostatin. A.B.K. and L.L.M. performed tissue location of transcripts analysis. A.J.W. and S.P.L. performed annotation of phosphatases. T.L.L., K.M.R., M.Mi., and R.C. performed annotation of actins and annotation of cardiac transcriptional program with D.L.T. D.J.J., B.K. and M.Me. performed annotation of biomineralization genes. J.C., A.K. and C.F. performed annotation of DNA transposons, global analysis of transposable element landscape, and horizontal transfer events. A.S and C.B. performed repeat/TE analysis. E.S.L., S.M.Z., G.M.M. and S.K.B. conducted comparative B. pfeifferi transcriptome studies and data analysis. C.M.A., did most of the writing with P.M., M.L.M. and contributions from all authors.

## Additional information

**Competing interests:** The authors declare no competing financial interests.

Coen M. Adema[1], LaDeana W. Hillier[2], Catherine S. Jones[3], Eric S. Loker[1], Matty Knight[4,5], Patrick Minx[2], Guilherme Oliveira[6,7], Nithya Raghavan[8], Andrew Shedlock[9], Laurence Rodrigues do Amaral[10], Halime D. Arican-Goktas[11], Juliana G. Assis[6], Elio Hideo Baba[6], Olga L. Baron[12], Christopher J. Bayne[13], Utibe Bickham-Wright[14], Kyle K. Biggar[15], Michael Blouin[13], Bryony C. Bonning[16], Chris Botka[17], Joanna M. Bridger[11], Katherine M. Buckley[18], Sarah K. Buddenborg[1], Roberta Lima Caldeira[19], Julia Carleton[20], Omar S. Carvalho[19], Maria G. Castillo[21], Iain W. Chalmers[22], Mikkel Christensens[23], Sandra Clifton[2], Celine Cosseau[24], Christine Coustau[12], Richard M. Cripps[25], Yesid Cuesta-Astroz[6], Scott F. Cummins[26], Leon Di Stefano[27,28], Nathalie Dinguirard[14], David Duval[24], Scott Emrich[29], Cédric Feschotte[20], Rene Feyereisen[30], Peter FitzGerald[31], Catrina Fronick[2], Lucinda Fulton[2], Richard Galinier[24], Sandra G. Gava[6], Michael Geusz[32], Kathrin K. Geyer[22], Gloria I. Giraldo-Calderón[29], Matheus de Souza Gomes[10], Michelle A. Gordy[33], Benjamin Gourbal[24], Christoph Grunau[24], Patrick C. Hanington[33], Karl F. Hoffmann[22], Daniel Hughes[23], Judith Humphries[34], Daniel J. Jackson[35], Liana K. Jannotti-Passos[6], Wander de Jesus Jeremias[6], Susan Jobling[36], Bishoy Kamel[37], Aurélie Kapusta[20], Satwant Kaur[36],

Joris M. Koene[38], Andrea B. Kohn[39], Dan Lawson[23], Scott P. Lawton[40], Di Liang[26], Yanin Limpanont[26], Sijun Liu[16], Anne E. Lockyer[36], TyAnna L. Lovato[25], Fernanda Ludolf[6], Vince Magrini[2], Donald P. McManus[41], Monica Medina[37], Milind Misra[1], Guillaume Mitta[24], Gerald M. Mkoji[42], Michael J. Montague[43], Cesar Montelongo[21], Leonid L. Moroz[39], Monica C. Munoz-Torres[44], Umar Niazi[22], Leslie R. Noble[3], Francislon S. Oliveira[6], Fabiano S. Pais[6], Anthony T. Papenfuss[27,28], Rob Peace[45], Janeth J. Pena[1], Emmanuel A. Pila[33], Titouan Quelais[24], Brian J. Raney[46], Jonathan P. Rast[18], David Rollinson[47], Izinara C. Rosse[6], Bronwyn Rotgans[26], Edwin J. Routledge[36], Kathryn M. Ryan[25], Larissa L.S. Scholte[6], Kenneth B. Storey[15], Martin Swain[22], Jacob A. Tennessen[13], Chad Tomlinson[2], Damian L. Trujillo[25], Emanuela V. Volpi[48], Anthony J. Walker[40], Tianfang Wang[26], Ittiprasert Wannaporn[4], Wesley C. Warren[2], Xiao-Jun Wu[14], Timothy P. Yoshino[14], Mohammed Yusuf[49,50], Si-Ming Zhang[1], Min Zhao[26] & Richard K. Wilson[2]

[1] Center for Theoretical and Evolutionary Immunology, Biology, University of New Mexico, Albuquerque, New Mexico 87131, USA. [2] The McDonnell Genome Institute, Washington University, Saint Louis, Missouri 63108, USA. [3] Institute of Biological and Environmental Sciences, School of Biological Sciences, University of Aberdeen, Tillydrone Avenue, Aberdeen AB24 2TZ, UK. [4] Department of Microbiology, Immunology & Tropical Medicine and Research Center for Neglected Diseases of Poverty, School of Medicine & Health Sciences, The George Washington University, Washington, District Of Columbia 20037, USA. [5] Division of Science & Mathematics, University of the District of Columbia, 4200 Connecticut Avenue NW Washington, Washington, District Of Columbia 20008, USA. [6] René Rachou Research Center, FIOCRUZ-Minas, Belo Horizonte, Minas Gerais 30190-002, Brazil. [7] Instituto Tecnológico Vale, Belém 66055-090, Brazil. [8] 10805 Tenbrook Dr Silver Spring, Maryland 20901, USA. [9] College of Charleston, Biology Department, Medical University of South Carolina College of Graduate Studies Hollings Marine Laboratory Charleston, Charleston, South Carolina 29412, USA. [10] Laboratory of Bioinformatics and Molecular Analysis, Institute of Genetics and Biochemistry Federal University of Uberlândia - Campus Patos de Minas (UFU), CEP 38700-128 Patos de Minas, Brasil. [11] Department of Life Sciences, College of Health and Life Sciences, Brunel University, London, Uxbridge UB8 3PH, UK. [12] Institut Sophia Agrobiotech, INRA/CNRS/UNS, Sophia Antipolis 06 903, France. [13] Department of Integrative Biology, Oregon State University, 3029 Cordley Hall, Corvallis, Oregon 97331, USA. [14] Department of Pathobiological Sciences, University of Wisconsin - School of Veterinary Medicine, 2015 Linden Dr., Madison, Wisconsin 53706, USA. [15] Institute of Biochemisty and Department of Biology, Carleton University, Ottawa, Ontario, Canada K1S 5B6. [16] Iowa State University, Ames, Iowa 50011, USA. [17] Department of Information Technology, Harvard Medical School, 107 Avenue Louis Pasteur, Boston, Massachusetts 02115, USA. [18] Sunnybrook Health Sciences Centre, Department of Immunology, University of Toronto, 2075 Bayview Avenue, Rm. S126, Toronto, Ontario, Canada M4N 3M5. [19] Laboratorio de Helmintologia e Malacologia Médica, FIOCRUZ-Minas, René Rachou Research Center, Belo Horizonte 30190-002, Brazil. [20] Department of Human Genetics, University of Utah. 15 North 2030 East, Salt Lake City, Utah 84112, USA. [21] Department of Biology, New Mexico State University, Las Cruces, New Mexico 88003, USA. [22] Animal and Microbial Sciences Research Theme, IBERS, Aberystwyth University, Aberystwyth SY23 3FG, UK. [23] EMBL-EBI, Wellcome Genome Campus, Hinxton, Cambridgeshire CB10 1SD, UK. [24] Univ. Perpignan Via Domitia, IHPE UMR 5244, CNRS, IFREMER, Univ. Montpellier, F-66860 Perpignan, France. [25] Department of Biology, University of New Mexico, Albuquerque, New Mexico 87131, USA. [26] Faculty of Science, Health and Education, University of the Sunshine Coast, Maroochydore, Queensland 4558, Australia. [27] The Walter and Eliza Hall Institute for Medical Research 1G Royal Parade, Parkville, Victoria 3052, Australia. [28] Bioinformatics and Cancer Genomics lab Lorenzo and Pamela Galli, Melanoma Research Fellow Peter MacCallum Cancer Centre St Andrews Place, East Melbourne, Victoria 3002, Australia. [29] Genomics and Bioinformatics Core Facility, 19 Galvin Life Sciences, University of Notre Dame, Notre Dame, Indiana 46556, USA. [30] University of Copenhagen, Faculty of Science, Department of Plant and Environmental Sciences, Thorvaldsensvej 40, 1871 Frederiksberg C, Denmark. [31] Genome Analysis Unit, National Cancer Institute, National Institutes of Health, Bethesda, Maryland 20892, USA. [32] Biological Sciences, Bowling Green State University, Bowling Green, Ohio 43403, USA. [33] Department of Public Health Sciences, University of Alberta, 3-57F South Academic Building, Edmonton, Alberta, Canada T6G 1C9. [34] Department of Biology, Lawrence University, Appleton, Wisconsin 54911, USA. [35] Courant Research Centre Geobiology, Georg-August University of Göttingen, Goldschmidtstraße 3, 37077 Göttingen, Germany. [36] Institute of Environment, Health & Societies, Environment and Health Theme, Brunel University London, Uxbridge UB8 3PH, UK. [37] Department of Biology, Pennsylvania State University, University Park, Pennsylvania 16802, USA. [38] Faculteit der Aard- en Levenswetenschappen, Vrije Universiteit, De Boelelaan 1085-1087, 1081 HV Amsterdam, The Netherlands. [39] The Whitney Laboratory for Marine Bioscience, University of Florida, 9505 Ocean Shore Blvd, St Augustine, Florida 32080, USA. [40] Molecular Parasitology Laboratory, School of Life Sciences Pharmacy and Chemistry, Kingston University, Kingston upon Thames, Surrey KT1 2EE, UK. [41] Molecular Parasitology Laboratory, QIMR Berghofer Medical Research Institute, Brisbane, Queensland 4006, Australia. [42] Kenya Medical Research Institute, P.O. Box 54840, 00200 Nairobi, Kenya. [43] Department of Neuroscience, Perelman School of Medicine at the University of Pennsylvania, Philadelphia, Pennsylvania 19104, USA. [44] Berkeley Bioinformatics Open-Source Projects, Environmental Genomics and Systems Biology Division, Lawrence Berkeley National Laboratory, One Cyclotron Road MS 977, Berkeley, California 94720, USA. [45] Department of Systems and Computer Engineering, Carleton University, Ottawa, Ontario, Canada K1S 5B6. [46] Genomics Institute, UC Santa Cruz, Santa Cruz, California 95064, USA. [47] Parasites and Vectors Division, London Centre for Neglected Tropical Disease Research, Wolfson Wellcome Biomedical Laboratories, Department of Life Sciences, Natural History Museum, Cromwell Road, London SW7 5BD, UK. [48] Department of Biomedical Sciences, Faculty of Science and Technology, University of Westminster, 115 New Cavendish Street, London W1W 6UW, UK. [49] London Centre for Nanotechnology, University College London, Gower Street, London WC1E 6BT, UK. [50] Research Complex at Harwell, Rutherford Appleton Laboratory, Oxfordshire OX11 0FA, UK.

DOI: 10.1038/ncomms16153    OPEN

# Corrigendum: Whole genome analysis of a schistosomiasis-transmitting freshwater snail

Coen M. Adema, LaDeana W. Hillier, Catherine S. Jones, Eric S. Loker, Matty Knight, Patrick Minx, Guilherme Oliveira, Nithya Raghavan, Andrew Shedlock, Laurence Rodrigues do Amaral, Halime D. Arican-Goktas, Juliana G. Assis, Elio Hideo Baba, Olga L. Baron, Christopher J. Bayne, Utibe Bickham-Wright, Kyle K. Biggar, Michael Blouin, Bryony C. Bonning, Chris Botka, Joanna M. Bridger, Katherine M. Buckley, Sarah K. Buddenborg, Roberta Lima Caldeira, Julia Carleton, Omar S. Carvalho, Maria G. Castillo, Iain W. Chalmers, Mikkel Christensens, Sandra Clifton, Celine Cosseau, Christine Coustau, Richard M. Cripps, Yesid Cuesta-Astroz, Scott F. Cummins, Leon Di Stefano, Nathalie Dinguirard, David Duval, Scott Emrich, Cédric Feschotte, Rene Feyereisen, Peter FitzGerald, Catrina Fronick, Lucinda Fulton, Richard Galinier, Sandra G. Gava, Michael Geusz, Kathrin K. Geyer, Gloria I. Giraldo-Calderón, Matheus de Souza Gomes, Michelle A. Gordy, Benjamin Gourbal, Christoph Grunau, Patrick C. Hanington, Karl F. Hoffmann, Daniel Hughes, Judith Humphries, Daniel J. Jackson, Liana K. Jannotti-Passos, Wander de Jesus Jeremias, Susan Jobling, Bishoy Kamel, Aurélie Kapusta, Satwant Kaur, Joris M. Koene, Andrea B. Kohn, Dan Lawson, Scott P. Lawton, Di Liang, Yanin Limpanont, Sijun Liu, Anne E. Lockyer, Ty Anna L. Lovato, Fernanda Ludolf, Vince Magrini, Donald P. McManus, Monica Medina, Milind Misra, Guillaume Mitta, Gerald M. Mkoji, Michael J. Montague, Cesar Montelongo, Leonid L. Moroz, Monica C. Munoz-Torres, Umar Niazi, Leslie R. Noble, Francislon S. Oliveira, Fabiano S. Pais, Anthony T. Papenfuss, Rob Peace, Janeth J. Pena, Emmanuel A. Pila, Titouan Quelais, Brian J. Raney, Jonathan P. Rast, David Rollinson, Izinara C. Rosse, Bronwyn Rotgans, Edwin J. Routledge, Kathryn M. Ryan, Larissa L.S. Scholte, Kenneth B. Storey, Martin Swain, Jacob A. Tennessen, Chad Tomlinson, Damian L. Trujillo, Emanuela V. Volpi, Anthony J. Walker, Tianfang Wang, Ittiprasert Wannaporn, Wesley C. Warren, Xiao-Jun Wu, Timothy P. Yoshino, Mohammed Yusuf, Si-Ming Zhang, Min Zhao & Richard K. Wilson

*Nature Communications* 8:15451 doi: 10.1038/ncomms15451 (2017); Published 16 May 2017; Updated 23 Aug 2017

The original version of this Article contained an error in the spelling of the author Leon Di Stefano, which was incorrectly given as Leon di Stephano. This has now been corrected in both the PDF and HTML versions of the Article.

