## [Peer Review File · Nature Communications]

Reviewers' comments:

Reviewer #1 (Remarks to the Author):

Title: Whole genome analysis of a schistosomiasis-transmitting freshwater snail

MS #: NCOMMS-16-04729-T

The paper marks a milestone of achievement for the Biomphalaria genome initiative and a triumph of international collaboration. Vectors of neglected tropical diseases remain largely understudied and the annotated genome of the vector of Schistosomiasis should begin revealing insights into multiple aspects of its biology and features relevant to host-pathogen interactions. I have no doubt that the work represents a massive effort and constitutes a significant contribution to the parasitology community. I do have some concerns, however.

One aspect that I found unsatisfying was the 'index' nature of the paper and the fact that the most interesting findings reside in the supplementary materials. Considering the space restriction that is presumably driving this structure, the Supplementary materials would benefit from a better organization and attention to rigor. In their current state, the methodology and results are rather difficult to assess and access and their format/presentation compromises the utility of this resource to the scientific community. I presume this is due to the fact that the supplementary material is predominantly organized as a collection of contributions from the various contributing groups.

In summary, a monumental achievement represented here in a paper that reads like a table of contents for tantalizing tidbits, leaving behind a rather dry core of listed observations and sending the reader frequently to a supplementary section that can benefit from more uniformity, and more organized content delivery. This is something that should be addressed considering the importance of this resource that transcends the short vignettes in the body of the paper.

Specific concerns

I found some aspects of the report are incomplete and crucial details are lacking. As a genomicist, I single out here a couple of components (Genome assembly and RNA-seq analyses) since they constitute a central resource and were selected for inclusion in the 'Methods' section. Other methodologies may be equally poorly developed but I am not qualified to review and assess all the other components.

The genome assembly should be better described including the source of all underlying reads and how to access them. A Table summarizing all type of reads collected, accessions for groups of reads (not the assembly), average read length, insert size, platform, mate pair status and clearly distinguishing clone coverage from sequence coverage. The data source table should allow a qualified reader to better evaluate this assembly, validate it, construct their own assembly or revise the existing one as more data become available in the future.

The RNA-seq data can offer a vast resource, yet it was only used to manually 'annotate' a subset of genes. Was an effort made to use the RNA-data to systematically and dramatically enhance the structural annotation of all predicted genes including exon-intron junctions, UTRs, etc...? Here also, an essential table is missing, summarizing the types of libraries constructed (PolyA+?), types of reads generated (SR vs PE), length, and coverage obtained for each library, along with SRA accessions.

Many of the analyses would benefit from a better organization. In their current state, the results difficult to access and their format/presentation compromises the utility of this resource to the scientific community. I presume this is due to the fact that the supplementary material is predominantly organized as a collection of contributions from the various contributing groups.

Here also, I use RNAseq analyses as an example, but the issues raised apply to multiple sections of the analyses. For the RNAseq analyses, the reader is sent to Supp. Text 3 (and related Figures and Tables). The section is called "Sequence mapping, variant calling, Eukaryotic protein kinase identification, Secretome prediction" and represents a concoction of analyses with no clear rationale of why they were grouped together. Many questions arise and are not answered: what is the significance of the variant calling? Where RNAs collected from a different strains of *B. glabrata*? Why are protein kinase analyses carried out on RNAseq data and not the full predicted gene models? The reader is provided some links to view analyses but the links are no longer valid (http://headnode.cebio.org/download/KAAS/B_glabrata/), etc...

Most of the supplementary tables and figures lack clarity and need legends to guide the reader through the content, but perhaps the most serious oversight is the lack of unified and consistent naming for all genes and genomic features reported in various analyses. This become quickly apparent when browsing through the supplementary tables where genes are listed using identifiers ranging from "BGLTMP000004-PA" to "Locus_4189_Transcript_8/8_Confidence_0.406_Length_7051" (Table S8) to "LG11_random_Scaffold4a" (Table S9).

Figures

I did not find Fig. 3 particularly informative. Am I missing the point? The large number of white boxes (indicating the mammalian proteins lacking an obvious ortholog in the snail) render most of the +p and -p arrows meaningless.

Fig 4 (ABC). The legend should provide greater detail, including the fact that the heatmaps were generated using FPKM values. Was the data log-transformed? If not where the negative score values coming from? Is the hierarchical clustering meaningful in this context?

The same applies to the D. panel. More detail is needed.

Minor comments

1. The authors state on line 75 that a linkage map was used to assign genomic contigs to linkage groups. In the Methods section, however, we're told that "Because of low marker density and (?) scaffolds could not be ordered and oriented within linkage groups."

2. Line 379. "We sequenced fragments (15X coverage), 3kb long inserts (10X), and 8kb long inserts (3X) with...". What are the fragments vs. the others inserts described? And certainly only end sequences were generated from all those inserts and not the full insert. This should be

Reviewer #2 (Remarks to the Author):

This paper describes the genome of a fresh-water lophotrochozoan *Biomphalaria glabrata*. While the numerous findings are in principle relevant to its evolution and biology, the presentation of the manuscript makes it hard for the reader to understand the actual insights. More than 200 page supplement is also very hard to read. It almost seems that the manuscript can be split up into several nicely written papers addressing different aspects of its biology with proper and sufficient description of the methods and caveats. For example, the general analysis of phero-perception, immune functions, pattern recognition etc does not necessarily (at least to me) relate to the (what seems to be) the main stories of parasite transmission, fresh water habitat adaptation, or even evolutionary peculiarities of the species, as those gene families generally tend to expand across lophotrochozoans. At the same time, potentially interesting insights coming directly from the genome comparison to other lophotrochozoans (such as marine to fresh water transition? gene novelties?) are left completely unaddressed or are described very briefly. To give an example from the very beginning: some promising results from secretome analysis are alluded to first, but then immediately forgotten once the topic is switched to GPCRs in the next sentence (in a rather hypothetical link). Then when discussing GPCRs the reader is left completely perplexed as to what those GPCRs exactly are, i.e., what classes, what does "GPCR-like" mean (is it a GPCR at all, why this particular gene)? Then, I am not an expert, but does 'anti-stress' response really consist only of HSPs? Further reading of the manuscript reveals several gene lists directly or marginally important for certain functions, but it's hard to either see why they are special or relevant to the story or the evidence/hypotheses that are provided are only briefly mentioned. For example, evolution of cardiac relevant genes comes out of nowhere. Figure 3 is left completely undescribed in the main text. This kind of superficial description of the findings is sometimes supplemented by the lack of a proper methods section. For example, there is no description how (using the example from above) GPCRs were identified and classified, same is true for neuropeptides or repetitive sequences, just to name a few. Annotation of all of those features requires specialized approaches/software that is not at all mentioned (or perhaps missed by me). In summary, such convoluted presentation of data makes it principally impossible to evaluate the findings and their importance. I would like to stress that I don't see any reason why the reported results are inaccurate and find them in principle interesting and actually relevant, but both the convoluted data presentation and the lack of proper description is quite worrying. Thus, in my view, this requires a complete re-write.

REVIEWERS' COMMENTS:

Reviewer #1 (Remarks to the Author):

The authors have adequately addressed my concerns and I continue to believe that this work represents an important and significant contribution to a broad audience of scientists working on pathogens and the vectors that transmit them.

Reviewer #2 (Remarks to the Author):

The reworked version of the manuscript has been significantly improved. I would like to thank the authors for their efforts in addressing reviewers' criticism and concerns. The manuscript is more streamlined and the analyses are now put in specific biologically-relevant context. Overall, the efforts that have been put into the analysis of this genome by the international consortium are now clearly visible. I have a couple of (minor) suggestions that I think would further improve this manuscript:

- It would help if the figures in the supplement would be attached to the individual notes and not come at the end of the document in a big lump
- "Additional to previous reports of *Capsaspora* ... novel agents that may find application in genetic modification or control of snails " To me this seems to be an over-interpretation of potential contaminants and I would suggest removing this sentence (or eluding to it in the Methods).
- "The HSP70 gene family is the largest with six multi-exon genes, five single exon genes, and over ten pseudogenes " My concern here is for the single exon genes - how complete are those gene models (or how many of them represent alleles of the better assembled copies)? Considering relative low N50 (48kb) for this genome, how do authors deal with such fragmented gene models? Generally, considering the relatively low mapping of transcripts to the genome (60%?), it would be good to know if any gene copies were missed in the other 40%.
- "Based on sequence similarities, several miRNAs likely regulate transcripts for processes unique to snail biology, including ..." Is this a quantitative statement, i.e., is there an underlying enrichment analysis (GO etc) for those functions or categories? The corresponding section 22 seems to be repeating itself yet somewhat eluding to such an analysis.
- While the genomic insights into species specific biology are generally well described and supported, I still find that the 'Bilaterian evolution' section lacks depth as presented in the current version of the manuscript. The conclusion of the proposed homology for a 'heart-like structure' in the urbilaterian ancestor does require a more complete expression and functional study. This problem is also not very well introduced. I find that the insights of this section (mainly, actin and biomineralization gene family analyses) can be integrated into the previous sections.
- Figure 4 - I do not quite understand the difference in the bar widths along the circle
- I am curious by the statement that "no intact transposases were detected in the assembly" (sic), yet there are 1000's of detectable hAT transposon copies. The supplement seems discordant with the main text as to which subfamily has been looked at. Have the authors checked for the presence of transposases in the transcriptomes? Generally speaking, the age curves (as shown in Figure 5) are misleading since they are heavily biased towards the way the consensus sequence was built.

Specific responses to reviewer comments,
Note that comments are shown in black, the responses are indicated in red

Reviewer #1 (Remarks to the Author):

One aspect that I found unsatisfying was the 'index' nature of the paper and the fact that the most interesting findings reside in the supplementary materials. Considering the space restriction that is presumably driving this structure, the Supplementary materials would benefit from a better organization and attention to rigor. In their current state, the methodology and results are rather difficult to assess and access and their format/presentation compromises the utility of this resource to the scientific community. I presume this is due to the fact that the supplementary material is predominantly organized as a collection of contributions from the various contributing groups. In summary, a monumental achievement represented here in a paper that reads like a table of contents for tantalizing tidbits, leaving behind a rather dry core of listed observations and sending the reader frequently to a supplementary section that can benefit from more uniformity, and more organized content delivery. This is something that should be addressed considering the importance of this resource that transcends the short vignettes in the body of the paper.

Response:

Indeed space restrictions are driving this "index" structure, however, sections of the main paper have been edited to make more accessible the intent of the analyses performed and the potential that is inherent to the findings. Additionally, we have followed the reviewer's recommendation for improving organization of the Supplementary materials and attention to uniformity and rigor. The supplementary information now includes opening background statements for the separate sections and methods for all supplementaries. The number of separate tables and figures has been reduced and legends were included with tables and figures.

Specific concerns

I found some aspects of the report are incomplete and crucial details are lacking. As a genomicist, I single out here a couple of components (Genome assembly and RNA-seq analyses) since they constitute a central resource and were selected for inclusion in the 'Methods' section. Other methodologies may be equally poorly developed but I am not qualified to review and assess all the other components.

Response:

In response to the general comment, every supplementary section has been evaluated and edited to improve methods and data presentation.

The genome assembly should be better described including the source of all underlying reads and how to access them. A Table summarizing all type of reads collected, accessions for groups of reads (not the assembly), average read length, insert size, platform, mate pair status and clearly distinguishing clone coverage from sequence coverage. The data source table should allow a qualified reader to better evaluate this assembly, validate it, construct their own assembly or revise the existing one as more data become available in the future.

Response: The methods section has been edited. Supplementary Table 1 was included to provide detailed information regarding the datasets, including accession numbers. The accession are also listed in the Data availability statement: The sequence data that support the findings of this study have been deposited in GenBank with the accession codes SRX005826-28; SRX008161-2; SRA480937; SRA480939; SRA480940; SRA480945; TI accessions 2091872204-2092480271; 2104228958-2104243968; 2110153721-2118515136; 2181062043-2181066224; 2193113537-2193116528; 2204642410-2204763511; 2204820860-2204852286; 2213009530-2213057324; 2260448774-2260450167; SRX648260-71. Also see (Supplementary Table 1.) The *Biomphalaria glabrata* genome project has been deposited at DDBJ/EMBL/GenBank under the accession number APKA00000000.1

The RNA-seq data can offer a vast resource, yet it was only used to manually 'annotate' a subset of genes. Was an effort made to use the RNA-data to systematically and dramatically enhance the structural annotation of all predicted genes including exon-intron junctions, UTRs, etc...?

Response:

The reviewer is accurate in evaluation of use of this resource. To clarify, the methods text states:

RNAseq data were mapped to the genome assembly (Supplementary note 3). No formal effort was made to use the RNA-data to systematically enhance the structural annotation. VectorBase did, however, make this RNAseq data available in WebApollo such that the community could use these data to correct exon-intron junctions, UTRs, etc. through community annotation. All of these community-based updates have been incorporated and are available via the current VectorBase gene set. Repeat features were analyzed and masked (Supplementary note 32; see Vectorbase *Biomphalaria-glabrata*-BB02_REPEATS.lib, *Biomphalaria-glabrata*-BB02_REPEATFEATURES_BglaB1.gff3.gz).

Here also, an essential table is missing, summarizing the types of libraries constructed (PolyA+?), types of reads generated (SR vs PE), length, and coverage obtained for each library, along with SRA accessions.

Response:

Supplementary Table 1 was included to provide detailed information regarding the datasets, including accession numbers. The accession are also listed in the Data availability statement: The sequence data that support the findings of this study have been deposited in GenBank with the accession codes SRX005826-28; SRX008161-2; SRA480937; SRA480939; SRA480940; SRA480945; TI accessions 2091872204-2092480271; 2104228958-2104243968; 2110153721-2118515136; 2181062043-2181066224; 2193113537-2193116528; 2204642410-2204763511; 2204820860-2204852286; 2213009530-2213057324; 2260448774-2260450167; SRX648260-71. Also see (Supplementary Table 1.) The *Biomphalaria glabrata* genome project has been deposited at DDBJ/EMBL/GenBank under the accession number APKA00000000.1

Many of the analyses would benefit from a better organization. In their current state, the results difficult to access and their format/presentation compromises the utility of this resource to the scientific community. I presume this is due to the fact that the supplementary material is predominantly organized as a collection of contributions from the various contributing groups.

Response:

Each of the analyses has been modified to achieve a more uniform, more comprehensive description of methods and results, including reduction of separate tables, legends for figures and tables, and use of uniform identifiers for sequences.

Here also, I use RNAseq analyses as an example, but the issues raised apply to multiple sections of the analyses. For the RNAseq analyses, the reader is sent to Supp. Text 3 (and related Figures and Tables). The section is called "Sequence mapping, variant calling, Eukaryotic protein kinase identification, Secretome prediction" and represents a concoction of analyses with no clear rationale of why they were grouped together.

Response:

The particular section 3 included several different analyses that were performed by one particular contributing research group. The different analyses built progressively on the fundamental, initial sequence mapping. The revised section 3 no longer includes the section on Eukaryotic protein kinase identification to simplify and make coherent the remaining content.

Many questions arise and are not answered: what is the significance of the variant calling?

Response:

The manuscript text was edited:

The pile up of reads revealed polymorphic transcripts (containing single nucleotide variants; SNV), that were correlated through KEGG¹⁰ analyses with metabolic pathways represented in the predicted proteome and the secretome (Supplementary Figs. 4-7), (Supplementary Table 7-8), (Supplementary Note 3). Combined with delineation of organ-specific patterns of gene expression (Supplementary Figs. 8,9), (Supplementary Table 9), (Supplementary Note 4), this provided potential molecular markers to help interpret *B. glabrata*'s responses to environmental insults and pathogens, including schistosome-susceptible mechanisms and resistant phenotypes. Supplementary note 3 now includes a background section that states:

RNASeq data revealed patterns of gene expression and aid gene modeling, as well as helped to derive sequence variation data by inspecting the pileup of RNAseq reads for synonymous and nonsynonymous variants along the expressed transcripts. The polymorphic genes were correlated to a diversity of metabolic pathways identified by KEGG analysis for the predicted proteome. Little is known about the diversity of secretome proteins overall, and regarding differential expression of secreted proteins from various *B. glabrata* tissues that could potentially interact with *S. mansoni* as these parasites develop within the snail host. This analysis was also performed for the subset of gene models that was predicted to represent secreted proteins. Secreted proteins are involved in vital biological processes such as cellular adhesion and migration, cell-cell communication, differentiation, proliferation and regulation of immune responses. Likely, these proteins are important for understanding host-parasite interactions. We predicted the whole set of secreted proteins, analyzed their diversity and annotated the putative secretome in terms of GO, Pfam domains and metabolic pathways.

Where RNAs collected from a different strains of *B. glabrata*?

Response:

Throughout the paper, sequence data (genomic and RNAseq) are most usually obtained from the BB02 strain, as described in the main methods section. Analyses that employed data from other snail strains now specifically state that in the relevant supplementary notes.

Why are protein kinase analyses carried out on RNAseq data and not the full predicted gene models?

Response:

The kinome identification was performed by using fasta sequences from the predicted proteome, not the RNASeq data. The whole set of 14,141 *B. glabrata* proteins was analyzed. Through hidden Markov model (HMM) searches potential homologs containing one of the diagnostic catalytic domains (PF00069 or PF07714) were selected

The reader is provided some links to view analyses but the links are no longer valid (http://headnode.cebio.org/download/KAAS/B_glabrata/), etc...

Response:

This matter has been fixed. We confirm that the data is available at http://headnode.cebio.org/download/KAAS/B_glabrata/

All links were been checked and updated for validity as needed

Most of the supplementary tables and figures lack clarity and need legends to guide the reader through the content, but perhaps the most serious oversight is the lack of unified and consistent naming for all genes and genomic features reported in various analyses. This become quickly apparent when browsing through the supplementary tables where genes are listed using identifiers ranging from "BGLTMP000004-PA" to "Locus_4189_Transcript_8/8_Confidence_0.406_Length_7051" (Table S8) to "" (Table S9).

Response:

Tables and figures have been evaluated and legends have been provided to guide the reader. Similarly, a unified and consistent naming for genes and genomic features was implemented. We adhere to identifiers implemented by Vectorbase, such that gene models are identified by e.g. BGLB000001 (with -RA or -PA designating expressed RNA or protein level sequence), scaffolds are identified e.g. as LG11_random_Scaffold4.

Regarding the example of LG11_random_Scaffold4a; a legend was included to indicate that letters identify consecutive genes that cluster on this scaffold, yet for which no gene model was available. For genes that currently lack a predicted gene model in the version of assembly used for this report, the sequence location is identified by scaffold ID and sequence interval, with a + or – for directionality, "LG11_random_Scaffold4:100-800,+". In case of use of other codes, a reference is made to the (URL) for on-line location of the relevant database, and activity of the URL was confirmed.

Figures

I did not find Fig. 3 particularly informative. Am I missing the point? The large number of white

boxes (indicating the mammalian proteins lacking an obvious ortholog in the snail render most of the +p and -p arrows meaningless.

Response:

Figure 3 has been removed

Fig 4 (ABC). The legend should provide greater detail, including the fact that the heatmaps were generated using FPKM values. Was the data log-transformed? If not where the negative score values coming from? Is the hierarchical clustering meaningful in this context? The same applies to the D. panel. More detail is needed.

Response:

Details are now provided in the legend and the relevant supplementary.

The use of FPKM values, calculated in standard manner (reference provided in Supplementary 31) is indicated. The hierarchical clustering has been removed. The legend now states: Figure 3. Expression of cardiac genes and actin genes in *B. glabrata* tissues. (A) Cardiac regulatory genes. (B) Cardiac structural genes. (C) Relative expression of actin genes in *B. glabrata* tissues. For A), B) and C), the score represents gene level aggregate of normalized FPKM counts for *de novo* assembled tissue transcripts, relative to expression levels in the heart/APO sample. The counts were scaled (with median read count as 0) to indicate expression intensity with red indicating highest, blue lowest. AG - Albumen gland; BUC - buccal mass; CNS - central nervous system; DG - digestive gland; FOOT – headfoot; HAPO - heart/APO; KID – kidney; MAN - mantle edge; OVO – ovotestes; SAL - salivary glands; STO - stomach; TRG - terminal genitalia (D) Maximum Likelihood tree (Phylogeny.fr, scale bar represents amino acid substitutions per site) showing phylogenetic relationships of actin genes, based on amino acid sequence alignment (ClustalW). *Biomphalaria* -snail; *Crassostrea gigas* – oyster; *Haliotis iris*– abalone; *Hirudo medicinalis* – leech (all lophotrochozoans); *Amphimedon queenslandica* - sponge, Prebilateria, ophotrochozoans), *Drosophila melanogaster* – fruit fly, Ecdysozoa), and the deuterostomes *Ciona intestinalis* - sea squirt; *Homo sapiens* – human. See Supplementary Note 31 for accession numbers.

Minor comments

1. The authors state on line 75 that a linkage map was used to assign genomic contigs to linkage groups. In the Methods section, however, we're told that "Because of low marker density and (?) scaffolds could not be ordered and oriented within linkage groups."

Response:

The line in question was corrected to "Because of low marker density, scaffolds could not be ordered and oriented within linkage groups." The latter statement refers that the notion that scaffolds were assigned to linkage groups, but that scaffolds within a linkage group were not placed in a specific order or direction.

2. Line 379. "We sequenced fragments (15X coverage), 3kb long inserts (10X), and 8kb long inserts (3X) with...". What are the fragments vs. the others inserts described? And certainly only end sequences were generated from all those inserts and not the full insert. This should be (...)

The information is provided in text and in tables now is as follows:

Using a genome size estimate of 0.9-1Gb⁷, we sequenced fragments (450bp read length; 14.08X coverage) and paired ends from 3kb long inserts (8.12X) and 8kb long inserts (2.82X) with reads generated on Roche 454 instrumentation, plus 0.06X from bacterial artificial chromosome (BAC) ends⁸ on the ABI3730xl. Reads were assembled using Newbler (v2.6)⁴⁵. Paired end reads from a 300bp insert library (53.42x coverage) were collected using Illumina instrumentation and assembled de novo using SOAP (v1.0.5)⁴⁶.

Supplementary table 1 provides additional details regarding read length for each of the libraries used.

Reviewer #2 (Remarks to the Author):

This paper describes the genome of a fresh-water lophotrochozoan *Biomphalaria glabrata*. While the numerous findings are in principle relevant to its evolution and biology, the presentation of the manuscript makes it hard for the reader to understand the actual insights.

Response:

The main manuscript has been revised for improved presentation, the following statements are included to clarify insights from the analyses performed :

- Combined with delineation of organ-specific patterns of gene expression (Supplementary Figs. 8,9), (Supplementary Table 9), (Supplementary Note 4), this provided potential molecular markers to help interpret *B. glabrata*'s responses to environmental insults and pathogens, including schistosome-susceptible mechanisms and resistant phenotypes
- Use of chemical communication systems to interact with conspecifics may have a tradeoff effect by potentially exposing *B. glabrata* as a target for parasites (Supplementary Figs. 10, 11), (Supplementary Table 11), (Supplementary Note 6) and that can be developed to interfere with snail mate finding and/or host location by parasites.
- Retention of HSP genes in *B. glabrata* embryonic (*Bge*) cells, the only available molluscan cell line¹⁴, enables *in vitro* investigation of anti-stress and pathogen responses involving *B. glabrata* HSPs
- These findings indicate potential for rational design of selective molluscicides, e.g. by inhibiting unique P450s or by activation of the molluscicide only by *B. glabrata*-specific P450s
- While gaps in functional annotation limit our interpretation of *B. glabrata* immune function (Supplementary Table 24,25), (Supplementary Note 19), our analyses reveal a multifaceted, complex internal defense system that must be evaded or negated by parasites such as *S. mansoni* to successfully establish infection.
- Characterization of the regulatory mechanisms that rule gene expression and general biological functions is especially interesting because survival of *B. glabrata* relies on the capacity to quickly recognize, respond, and adapt to external and internal signals. Additionally, a better understanding of parasite-host compatibility will be afforded by characterization of snail control mechanisms for gene expression and signaling pathways as possible targets for interference by *S. mansoni* to alter host physiology, including reproductive activities, in order to survive in *B. glabrata*
- Based on sequence similarities, several miRNAs likely regulate transcripts for processes unique to snail biology, including secretory mucosal proteins and shell formation that may present possible targets for control of *B. glabrata* (Supplementary Figs. 40-67), (Supplementary Tables 28-33), (Supplementary Note 21,22).

- Modification of expression of clock genes may interrupt circadian rhythms of *B. glabrata* and affect feeding, egg-laying and emergence of cercariae (Supplementary Note 23).
- Neuropeptides expressed within the nervous system coordinate the complex physiology of *B. glabrata*, a simultaneous hermaphrodite snail.
- Putatively, MAGPs evolve rapidly and are taxon-specific (Supplementary Fig. 69), (Supplementary Table 34), (Supplementary Note 25), such that they allow for specific targeting of reproductive activity for control measures.
- Characterization of snail-specific aspects of steroidogenesis may identify targets to disrupt reproduction toward control of snails. (Supplementary Fig. 70), (Supplementary Table 37), (Supplementary Note 26).
- Eukaryotic protein kinases (ePKs) and phosphatases constitute the core of cellular signaling pathways, playing a central role in signal transduction by catalyzing reversible protein phosphorylation in non-linearly integrated networks. mediate signal transduction through phosphorylation in complex networks, and protein phosphatases counteract these effects towards effective signaling. *Schistosoma mansoni* likely interferes with the extracellular signal-regulated kinase (ERK) pathway to survive in *B. glabrata*. These sequences can be studied for understanding control of homeostasis, particularly in the face of environmental and pathogenic insults encountered by *B. glabrata*.
- We found further support for the hypothesis, based on previous consideration of only ecdysozoans and deuterostomes, that a primitive heart-like structure which developed through the actions of a core heart toolkit, was present in the urbilaterian ancestor
- One interpretation is that actin genes diverged independently multiple times in molluscs, similar to an earlier hypothesis for independent actin diversification in arthropods and chordates³⁹. Alternatively, a stronger appearance of monophyly than really exists may result if selective pressures due to functional constraints keep actin sequences similar within a genome, for example if the encoded proteins have overlapping functions
- Highly conserved components of the molluscan shell forming toolkit include carbonic anhydrases and tyrosinases
- Overall, our results reinforce a model in which diverse repeats comprise a large fraction of molluscan genomes
- At least 90% sequence identity was shared among 196 assembled transcripts collected from *B. pfeifferi* (Illumina RNAseq) and the transcriptome of *B. glabrata* (Supplementary Tables 42-43), (Supplementary Note 33). This report provides novel details on the biological properties of *B. glabrata*, including several that may help determine suitability of *B. glabrata* as intermediate host for *S. mansoni*, and points to potential approaches for more effective control efforts against *Biomphalaria* to limit the transmission of schistosomiasis.

More than 200 page supplement is also very hard to read. It almost seems that the manuscript can be split up into several nicely written papers addressing different aspects of its biology with proper and sufficient description of the methods and caveats.

Response:

The supplementaries have been edited extensively to improve readability. The volume of the supplementary information is defined by the work from the collection of research groups that have contributed their work to this effort. The numbers of both figures and tables have been reduced as possible.

For example, the general analysis of phero-perception, immune functions, pattern recognition, etc does not necessarily (at least to me) relate to the (what seems to be) the main stories of parasite transmission, fresh water habitat adaptation, or even evolutionary peculiarities of the species, as those gene families generally tend to expand across lophotrochozoans.

Response:

However, specific to this organism as intermediate host for schistosome transmission. Communication in water relates to persistence in environment and mate finding, immune function is important because schistosomes defeat host immunity and it is unclear what the resistance genes are, especially how nonself recognition can be defeated. The sequences proper are relevant as targets for future functional analysis. They relate to parasite transmission this has been clarified more, and less critically so to adaptations and evolutionary peculiarities.

At the same time, potentially interesting insights coming directly from the genome comparison to other lophotrochozoans (such as marine to fresh water transition? gene novelties?) are left completely unaddressed or are described very briefly.

Response:

This genome project was funded by NHGRI specifically to focus on the biology of *B. glabrata* that was deemed important for transmission of schistosomiasis. This general topic is also the scientific expertise of many of the co-authors that accepted the invitation to take part in this first analysis of the *Biomphalaria* genome. Some broader topics relating to Bilaterian evolution (heart development, actin evolution, biomineralization, Repetitive landscape) were included to utilize *Biomphalaria* to inform on general biology also. With the public availability of the sequence data, additional topics such as listed by the reviewer can be analyzed by others in future research.

To give an example from the very beginning: some promising results from secretome analysis are alluded to first, but then immediately forgotten once the topic is switched to GPCRs in the next sentence (in a rather hypothetical link). Then when discussing GPCRs the reader is left completely perplexed as to what those GPCRs exactly are, i.e., what classes, what does "GPCR-like" mean (is it a GPCR at all, why this particular gene)?

Response:

These topics are considered rather as components of two sides of previously not characterized communication capabilities of *B. glabrata* that have likely relevance for snail mate finding and for location of snail intermediate hosts by the parasite. Details and literature references regarding characterizations, identification and annotation/classification of GPCRs are now included in the supplementary note 6..

Then, I am not an expert, but does 'anti-stress' response really consist only of HSPs?

Response:

Indeed, stress responses are more diverse and can consist of more than HSPs. Our analyses have focused on HSP and CYPs, however. The text has been edited to state:
Five families of heat shock proteins (HSP): HSP20, HSP40, HSP60, HSP70, and HSP90 **contribute** to anti-stress response capabilities of *B. glabrata*. Additionally, *B. glabrata* has about 99 genes encoding heme-thiolate enzymes (CYP superfamily) toward detoxifying xenobiotics, with representation of all major animal cytochrome P450 clans.

Further reading of the manuscript reveals several gene lists directly or marginally important for certain functions, but it's hard to either see why they are special or relevant to the story or the evidence/hypotheses that are provided are only briefly mentioned. For example, evolution of cardiac relevant genes comes out of nowhere.

The manuscript was edited to introduce the evolution of cardiac relevant genes and other aspects of metazoan biology as follows:

Bilaterian evolution

Genome analyses of *B. glabrata* also provide insights into evolution of bilaterian metazoan by increasing diversity of the relatively few lophotrochozoan taxa that have been characterized to date (i.e. platyhelminths, leech, bivalve, cephalopod, polychaete), and through comparison with other animal clades.

Figure 3 is left completely undescribed in the main text.

Response:

Figure 3 has been deleted

This kind of superficial description of the findings is sometimes supplemented by the lack of a proper methods section. For example, there is no description how (using the example from above) GPCRs were identified and classified, same is true for neuropeptides or repetitive sequences, just to name a few. Annotation of all of those features requires specialized approaches/software that is not at all mentioned (or perhaps missed by me).

Response:

This valid comment has been addressed in detail by extensively editing the supplementary information to include methods sections (providing methods and relevant references) in the supplementary notes.

In summary, such convoluted presentation of data makes it principally impossible to evaluate the findings and their importance. I would like to stress that I don't see any reason why the reported results are inaccurate and find them in principle interesting and actually relevant, but both the convoluted data presentation and the lack of proper description is quite worrying. Thus, in my view, this requires a complete re-write.

Response:

As detailed above in the responses to specific comments, input from all contributing authors has been applied to effect extensive editing of all components of this complex report in order to improve data presentation and provide proper descriptions of the analyses performed. The text has been edited to clarify the findings and the importance of novel details regarding the biological properties of *B. glabrata*, including several that may help determine suitability of *B. glabrata* as intermediate host for *S. mansoni*, and that point to potential approaches for more effective control efforts against *Biomphalaria* to limit the transmission of schistosomiasis.

Your reference NCOMMS-16-04729

Dear Editor,

Listed below is the point-by-point response to comments from two reviewers regarding the edited manuscript (NCOMMS-16-04729): Whole genome analysis of a schistosomiasis-transmitting freshwater snail, by Adema et al., (117 authors).

We thank the reviewers for their constructive comments. We have also followed you editorial comments in developing our responses.

Comments and responses follow:

Your editorial comment: 2.1/ Please address the remaining concerns of referee 2 with textual revision. While formal re-review with this referee will not be necessary, other editors and I will check the final revision in order to make sure that these minor concerns are met with sufficient changes to the paper.

REVIEWERS' COMMENTS:

REVIEWER #1 (REMARKS TO THE AUTHOR):

The authors have adequately addressed my concerns and I continue to believe that this work represents an important and significant contribution to a broad audience of scientists working on pathogens and the vectors that transmit them.

RESPONSE We thank the reviewer for constructive comments that have helped improve our manuscript.

REVIEWER #2 (REMARKS TO THE AUTHOR):

The reworked version of the manuscript has been significantly improved. I would like to thank the authors for their efforts in addressing reviewers' criticism and concerns. The manuscript is more streamlined and the analyses are now put in specific biologically-relevant context. Overall, the efforts that have been put into the analysis of this genome by the international consortium are now clearly visible. I have a couple of (minor) suggestions that I think would further improve this manuscript:

GENERAL RESPONSE: We thank the reviewer for constructive comments that helped to improve the manuscript. For each current comment/question (Q), the answer (A) detailing the textual revisions are listed below.

Q - It would help if the figures in the supplement would be attached to the individual notes and not come at the end of the document in a big lump.

A - The supplement has been edited according to editorial instructions in Comment MC22: Supplementary Information Order: Figures, (tables), notes, references. Tables are included in the Supplementary Data.

Q - "Additional to previous reports of Capsaspora ... novel agents that may find application in genetic modification or control of snails " To me this seems to be an over-interpretation of potential contaminants and I would suggest removing this sentence (or eluding to it in the Methods).

A – The text has been modified to include a caveat regarding the need for verification of the biological reality and implication of these novel agents.

The revised text states: Pending further characterization of prevalence, specificity of association with *B. glabrata*, and impact on snail biology, these novel agents may find application in genetic modification of *B. glabrata* or control of snails through use of natural pathogens

Q - "The HSP70 gene family is the largest with six multi-exon genes, five single exon genes, and over ten pseudogenes " My concern here is for the single exon genes - how complete are those gene models (or how many of them represent alleles of the better assembled copies)? Considering relative low N50 (48kb) for this genome, how do authors deal with such fragmented gene models? Generally, considering the relatively low mapping of transcripts to the genome (60%?), it would be good to know if any gene copies were missed in the other 40%.

A- the existence of single exon HSP70 genes was independently confirmed by full length sequence of a BAC insert, to address this matter and the other concerns raised by the reviewer the text now reads: In general, it is anticipated that future genome assemblies and continued annotation efforts can identify additional *B. glabrata* genes and provide updated gene models to reveal that some current pseudogenes are in fact intact functional genes. The existence of a single exon HSP70 gene, however, was independently confirmed by sequence obtained from *B. glabrata* BAC clone (BG_BB a-117G16, Genbank AC233578, basepair interval 49686-51425), and this supports the notion that prediction of single exon gene models for several HSP70 genes from the current genome assembly is accurate.

Q - "Based on sequence similarities, several miRNAs likely regulate transcripts for processes unique to snail biology, including ..." Is this a quantitative statement, i.e., is there an underlying enrichment analysis (GO etc) for those functions or categories? The corresponding section 22 seems to be repeating itself yet somewhat eluding to such an analysis.

A – The text has been updated for clarification: Bioinformatics predicted 107 novel pre-miRNAs unique to *B. glabrata*. Based on the analysis of binding thermodynamics and miRNA:mRNA structural features, several novel miRNAs were predicted to likely regulate transcripts involved in processes unique to snail biology, including secretory mucosal proteins and shell formation (biomineralization) that may present possible targets for control of *B. glabrata* (Supplementary Figs. 40-67), (Supplementary Note 21,22), (Supplementary Data 28-33),

Q - While the genomic insights into species specific biology are generally well described and supported, I still find that the 'Bilaterian evolution' section lacks depth as presented in the current version of the manuscript. The conclusion of the proposed homology for a 'heart-like structure' in the urbilaterian ancestor does require a more complete expression and functional study. This problem is also not very well introduced. I find that the insights of this section (mainly, actin and biomineralization gene family analyses) can be integrated into the previous sections.

A – we have incorporated the following text to better introduce the section on bilateral evolution and present our conclusions carefully. The text now reads: Genome study of *B. glabrata* can also provide new insights into evolution of bilaterian metazoa by increasing diversity of the relatively few lophotrochozoan taxa that have been characterized to date (i.e. platyhelminths, leech, bivalve, cephalopod, polychaete)³²⁻³⁶. . Comparison of similar biological features and gene expression patterns among lophotrochozoans, ecdysozoans and deuterostomes may indicate the evolutionary origin of conserved gene families and anatomical. The prevalence in diverse taxa of metazoa, including molluscs, arthropods and chordates, of muscular heart-like organs that function to circulate blood or hemolymph, has led to the proposal that these structures evolved over evolutionary time from a primitive heart present in an urbilaterian ancestor. This hypothesis is supported by similarities in core genes for specification and differentiation of cardiac structures between insects (in particular *Drosophila*) and vertebrates^{37,38}. To further develop this notion, we searched for molluscan cardiac-specification.....

..... Pending confirmation of functional involvement of these core cardiac genes in heart formation , these results from a lophotrochozoan, in conjunction with ecdysozoans and deuterostomes, merit continued consideration of the presence of a primitive heart-like structure and in the urbilaterian ancestor.

We also investigated in molluscs, relative to insects and mammals, the evolution of the gene family of actins, conserved proteins that function in cell motility (cytoplasmic actins) and muscle contraction (sarcomeric actins)⁴⁰. Previous study showed that cephalopod actin genes⁴¹, are more closely related to one another than to any single mammalian gene, an observation also made another mollusc *Haliotis*⁴² and for insect actins⁴³. Thus, it has been proposed that actin diversification in arthropods, mollusks and vertebrates each occurred independently. However, it has not been determined if different molluscan lineages independently underwent actin gene divergence, and few studies have analyzed expression of mollusk actin genes in different tissues^{42,44}. We identified ten actin genes in *B. glabrata* that are clustered

..... To gain insight into the diversification of mechanisms involved in biomineralization in molluscs, we analyzed the transcriptomic data.....

..... In summary, this genome-level analysis of a subset of molluscan molecular pathways provides new insight into the evolutionary origins of bilaterian organs, gene families, and genetic pathways.

Q - Figure 4 - I do not quite understand the difference in the bar widths along the circle - I am curious by the statement that "no intact transposases were detected in the assembly" (sic), yet there are 1000's of detectable hAT transposon copies. The supplement seems discordant with the main text as to which subfamily has been looked at. Have the authors

checked for the presence of transposases in the transcriptomes? Generally speaking, the age curves (as shown in Figure 5) are misleading since they are heavily biased towards the way the consensus sequence was built.

A – :Figure 4” The different bar widths along the circle is explained in the legend. The width of each sector line around the ideogram is proportional to the length of that gene in base pairs,

A- regarding “the I am curious....sequence was built”:

There is a difference between intact transposase and recognizable transposase; the SPIN elements (hAT mentioned here) detected were all non-autonomous. The team looked with special attention for intact transposases, but observed a lot of stop codons where the copies were showing homology to the coding region of a transposase. The statement about no intact transposase in the assembly is true, however, this indeed does not mean that there are none in the non assembled genome. We did not inspect the transcriptomes (assembled and raw reads) for complete transposases because such analyses would take considerable time, considering the size of the dataset. We consider Figure 5 a great representation of the TE landscape but address the concern from the reviewer by including a note of caution in the figure legend : Note that the result of this analysis of assembled sequence does not exclude the likelihood that intact transposable elements are present in *B. glabrata*.